# $\phi$-Balancing for Mixture-of-Experts Training

**Lizhang Chen** [* 1]   **Jonathan Li** [* 1]   **Qi Wang** [* 1]
**Runlong Liao** [1]   **Shuozhe Li** [1]   **Chen Liang** [2]   **Ni Lao** [1]   **Qiang Liu** [1]

## Abstract

Mixture-of-Experts (MoE) models rely on balanced expert utilization to fully realize their scalability. However, existing load-balancing methods are largely heuristic and operate on noisy mini-batch assignment statistics, introducing bias relative to population-level objectives. We propose $\phi$-**balancing**, a principled framework that directly targets population-level expert balance by minimizing a strictly convex, symmetric, and differentiable potential of the expected routing distribution. Using convex duality, we derive an equivalent min-max formulation and obtain a simple online algorithm via mirror descent, yielding an efficient EMA-based routing adjustment with negligible overhead. Across large-scale pretraining and downstream fine-tuning, $\phi$-balancing consistently outperforms prior Switch-style and loss-free baselines, demonstrating more stable and effective expert utilization.

## 1. Introduction

Mixture-of-Experts (MoE) Transformers have emerged as an effective approach for scaling deep learning models by dynamically selecting a small subset of expert modules for each input token. This strategy substantially increases model capacity while keeping computation nearly constant (Shazeer et al., 2017; Fedus et al., 2022), enabling large-scale language and vision models with billions of parameters to operate at roughly constant FLOPs (Lepikhin et al., 2021; Riquelme et al., 2021; Fedus et al., 2022).

A key challenge in MoE training is to ensure balanced utilization of experts, which is essential for fully leveraging model capacity and avoiding performance degradation. A number of methods have been proposed to ad-

---

**Algorithm 1** $\phi$-balancing for one MoE layer

**Require:** strictly convex, symmetric, and differentiable $\phi$, $\eta \in (0, 1]$, $\alpha > 0$, $\mathbf{m} \leftarrow \mathbf{0}$, routing frequencies $f_e$ (4)
1: Compute routing probabilities $p_{i,e}$ for each token $i$
2: $\mathbf{p}_e \leftarrow \frac{1}{T}\sum_{i=1}^{T} p_{i,e}$ for $e = 1, \dots, E$     (expert loads)
3: Let $\mathbf{p} = (\mathbf{p}_1, \dots, \mathbf{p}_E)$
4: $\mathbf{m} \leftarrow (1 - \eta)\mathbf{m} + \eta\mathbf{p}$                    (EMA of loads)
5: $\mathcal{L}_{\text{aux}} \leftarrow \begin{cases} \textbf{ST-MoE: } \sum_{e=1}^{E} f_e\mathbf{p}_e \\[1em] \textbf{Ours: } \sum_{e=1}^{E} \nabla\phi(\mathbf{m})_e\mathbf{p}_e \end{cases}$
6: Update model using $\nabla\big(\mathcal{L}_{\text{task}} + \alpha \cdot E \cdot \mathcal{L}_{\text{aux}}\big)$

---

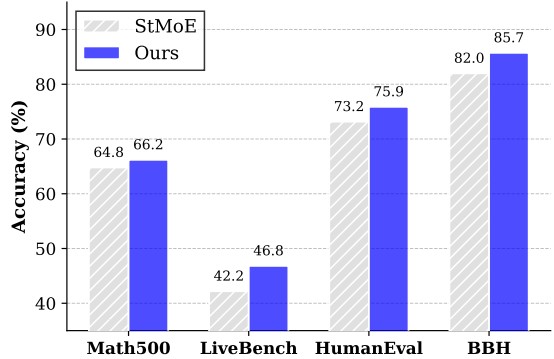

*Figure 1.* **Performance gains on reasoning and code generation benchmarks.** We compare the proposed method (*Ours*) against the ST-MoE baseline on the Moonlight-16B-A3B-Instruct architecture (Liu et al., 2025). The proposed approach outperforms the baseline across all selected tasks, yielding significant gains in mathematical reasoning (Math500), general capability (LiveBench), code synthesis (HumanEval), and logic (BBH).

dress this challenge, including Switch-style load-balancing losses (Shazeer et al., 2017; Lepikhin et al., 2021; Fedus et al., 2022) and more recent loss-free balancing approaches (Wang et al., 2024). However, an often unspoken issue is that most existing balancing objectives are heuristic in nature, as they do not correspond to minimizing a well-defined population-level objective. In principle, the true goal is to achieve balanced expert usage under the whole data distribution. In contrast, widely used methods such as Switch-style MoE (ST-MoE) rely on per-mini-batch statistics and realized

---

[*]Equal contribution   [1]University of Texas at Austin   [2]Northwestern University.   Correspondence to: Ni Lao <nlao@utexas.edu>, Qiang Liu <lqiang@cs.utexas.edu>.

*Proceedings of the $43^{rd}$ International Conference on Machine Learning*, Seoul, South Korea. PMLR 306, 2026. Copyright 2026 by the author(s).

expert assignment frequencies, which introduce systematic bias relative to population-level uniformity objectives.

We propose $\phi$-balancing, a principled load-balancing framework that directly targets population-level expert balance. Our approach formulates load balancing as the minimization of a strictly convex, symmetric, and differentiable potential $\phi$ applied to the population mean routing distribution. To avoid the bias introduced by per-batch approximation, we adopt a min-max formulation via convex duality and apply online mirror descent to solve the resulting inner problem. This yields a simple yet broad family of algorithms, shown in Algorithm 1, that maintains an exponential moving average (EMA) of routing probabilities with negligible overhead, processed through the mirror map $\nabla\phi$.

Empirically, we find that $\phi$-balancing consistently outperforms ST-MoE across a wide range of settings (Figure 1), including pretraining MoE-augmented Gemma models (Kamath et al., 2025; Liang et al., 2025), where we systematically scale the number of active parameters $N$, expert count $E$, and routing granularity $G$ under controlled compute budgets, and ablations on EMA-based load tracking, the choice of mirror map $\phi$, and the EMA decay rate. While many choices for $\phi$ are possible, we recommend the negative entropy function as the most effective in practice.

We further evaluate per-benchmark LoRA fine-tuning on instruction-tuned MoE backbones (DeepSeek-AI, 2024b; Dai et al., 2024; Liu et al., 2025) across seven benchmarks, totaling approximately 40,000 NVIDIA H100 HBM3-80GB GPU hours for all experiments.

## 2. Background on Mixtures of Experts

We consider a standard decoder-only Transformer composed of $L$ layers. In a dense Transformer, each layer processes the input sequence via a Self-Attention module followed by a shared Feed-Forward Network (FFN). The MoE architecture replaces this dense FFN with a sparse modular layer consisting of a learnable router and a set of $E$ experts, $\{\text{FFN}_1, \ldots, \text{FFN}_E\}$ (Shazeer et al., 2017).

Let $\boldsymbol{x} = (\boldsymbol{x}_i)_{i=1}^T \in \mathbb{R}^{T \times d}$ denote the input to a layer, where $T$ is the sequence length and $d$ is the model hidden dimension. For each token $\boldsymbol{x}_i$, the MoE layer output $\boldsymbol{y}_i$ is computed as the router-weighted sum of the experts:

$$\boldsymbol{y}_i = \sum_{e=1}^E R(\boldsymbol{x}_i)_e \cdot \text{FFN}_e(\boldsymbol{x}_i; d_{\text{ffn}}). \quad (1)$$

Here, each expert is parameterized as a standard two-layer MLP. Following recent state-of-the-art implementations (Shazeer, 2020; Dai et al., 2024; OpenAI, 2025), we utilize the SwiGLU activation function, defined as:

$$\text{FFN}_e(\boldsymbol{u}) = W_2^{(e)} \cdot \text{SwiGLU}(W_1^{(e)}\boldsymbol{u}), \quad (2)$$

where $W_1^{(e)} \in \mathbb{R}^{d_{\text{ffn}} \times d}$ and $W_2^{(e)} \in \mathbb{R}^{d \times d_{\text{ffn}}}$ are independent parameters for expert $e$.

### 2.1. Sparse Routing Mechanism

The computational efficiency of MoEs relies on the routing function $R(\cdot)$, which enforces sparsity by directing each token to a small subset of $k$ experts (where $k \ll E$). The router typically consists of a learnable projection matrix $W_r \in \mathbb{R}^{E \times d}$. The *routing weights* are determined by normalizing the projection scores over the top-$k$ indices (Shazeer et al., 2017):

$$R(\boldsymbol{x}) = \text{softmax}\left(\text{Top-}k(W_r\boldsymbol{x})\right). \quad (3)$$

The Top-$k(\cdot)$ operator sets all logits to $-\infty$ except for the $k$ largest elements. Consequently, $R(\boldsymbol{x})_e$ is zero for all non-selected experts, allowing the model to skip the majority of expert computations. If we only have one activated expert, then we will not do *softmax* to avoid zero gradient on the router logits. This conditional computation decouples parameter count from inference cost; however, it introduces the load-balancing challenges that we address in Section 3.

### 2.2. Baseline Load Balancing Strategy

While the router constitutes a negligible fraction of the total parameter count, it orchestrates the utilization of the model's vast expert capacity. Here, we recall the standard auxiliary load-balancing loss (LBL) used by ST-MoE (Fedus et al., 2022). This formulation remains the dominant paradigm for training large-scale sparse models, including DeepSeek (DeepSeek-AI, 2024b), OlMoE (Muennighoff et al., 2025), and DeepSeed-MoE (Rajbhandari et al., 2022).

The LBL objective encourages tokens to be distributed uniformly across the $E$ experts. For a minibatch of $T$ tokens, let $\mathbf{p}_e$ denote the batch-mean *pre-top-$k$* routing probability assigned to expert $e$, let $p_{i,e}$ denote the routing probability of expert $e$ for token $\boldsymbol{x}_i$, and let $f_e$ denote the realized routing frequency of expert $e$ under top-$k$ routing:

$$\mathbf{p}_e = \frac{1}{T}\sum_{i=1}^T p_{i,e}, \quad \text{where } p_{i,e} := \text{softmax}(W_r\boldsymbol{x}_i)_e,$$
$$f_e = \frac{1}{kT}\sum_{i=1}^T \mathbb{I}(e \in \text{Top-}k(W_r\boldsymbol{x}_i)). \quad (4)$$

The auxiliary loss is defined as the dot product of these two vectors:

$$\mathcal{L}_{\text{aux}} = \sum_{e=1}^E f_e \cdot \mathbf{p}_e. \quad (5)$$

As shown by Fedus et al. (2022), minimizing (5) encourages both the gating probabilities and the discrete selections to approach a uniform distribution.

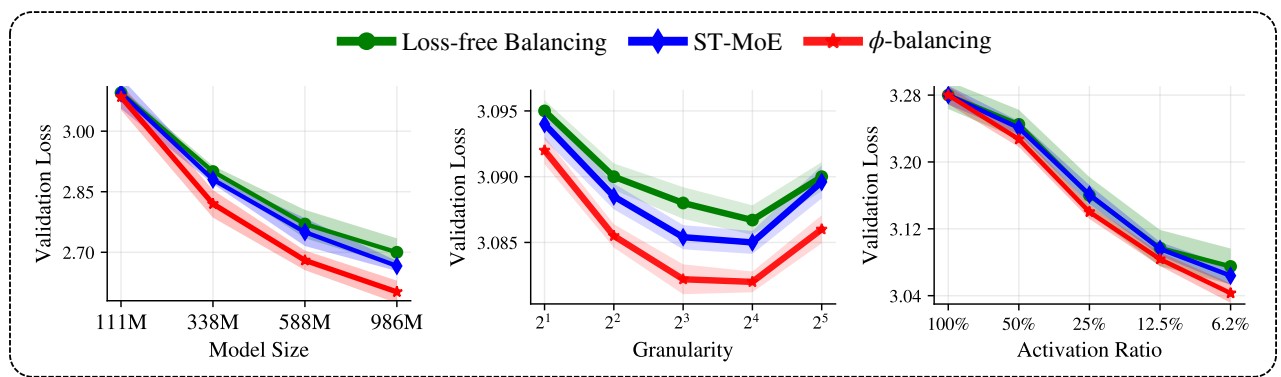

*Figure 2.* **Pretraining scaling studies under controlled per-token compute.** We evaluate routing stability and optimization across three orthogonal MoE scaling axes, while keeping the per-token computational cost (FLOPs) approximately constant within each study by adjusting expert size as needed. **(Left) Active-parameter scaling:** we train models with $E = 16$ experts and $A = 2$ active experts per token, varying the number of *active parameters* $N \in \{111M, 338M, 588M, 986M\}$. **(Middle) Granularity scaling:** for fixed model size $M$ and activation ratio $A/E$, we vary the granularity factor $G \in \{2, 4, 8, 16, 32\}$ by increasing the total number of experts from 16 to 256 while proportionally shrinking each expert, so per-token FLOPs remain constant. **(Right) Expert-count scaling (activation ratio):** we isolate the effect of $A/E$ by holding the compute budget $M$, the number of activated experts $A = 2$, and the expert size (granularity) fixed, and varying the total number of experts $E \in \{8, 16, 32, 64, 128\}$.

**Loss-free balancing.** Rather than introducing an explicit load-balancing loss, which can inject interference gradients and degrade task learning, *loss-free balancing* (Wang et al., 2024) enforces balance by directly modifying the routing decision. Concretely, it adds a learned, expert-specific bias to the router logits *before* the top-$k$ selection, and updates these biases online using each expert's recent utilization.

## 3. $\phi$-balancing

In this section, we introduce the $\phi$-balancing loss. Unlike classical approaches that enforce balance only within individual mini-batches, our goal is to regularize *global* expert usage over the entire data distribution. Concretely, we encourage globally uniform expert utilization via a strictly convex, symmetric, and differentiable potential function $\phi$.

### 3.1. The Global Load-Balancing Objective

Let $\mathbf{p}(\boldsymbol{x}; \theta) \in \Delta^E$ denote the predicted routing probability vector for an input token $x$, parameterized by $\theta$ (i.e., $\mathbf{p}(\boldsymbol{x}; \theta) = \text{softmax}(W_r \boldsymbol{x})$). For a specific expert $e$, $\mathbf{p}(\boldsymbol{x}; \theta)_e$ represents the probability mass assigned to that expert. We define the *global mean routing distribution* $\bar{\mathbf{p}}(\theta)$ as the expectation of the routing probabilities over the distribution of tokens $\mathcal{D}$ induced by the training corpus:

$$\bar{\mathbf{p}}(\theta) = \mathbb{E}_{\boldsymbol{x} \sim \mathcal{D}} \left[ \mathbf{p}(\boldsymbol{x}; \theta) \right], \tag{6}$$

which satisfies $\sum_{e=1}^{E} \bar{\mathbf{p}}(\theta)_e = 1$.

**Load balancing via convex duality.** Our goal is to encourage the token population-level routing distribution $\bar{\mathbf{p}}(\theta)$ to be uniform, so that in expectation, all experts are utilized equally over the data distribution. We formulate this

objective as the optimization problem

$$\min_{\theta} \mathcal{L}_{\text{bal}}(\theta) := \min_{\theta} \phi(\bar{\mathbf{p}}(\theta)), \tag{7}$$

where the potential function $\phi : \mathbb{R}^E \to \mathbb{R}$ is chosen to be strictly convex, symmetric, and differentiable.

The strict convexity and symmetry of $\phi$ guarantee that the objective in (7) attains a unique minimum over the probability simplex at the uniform distribution, which is formalized by Lemma 1 in Appendix B. Representative choices of $\phi$ are summarized in Table 1. Importantly, $\phi$ is not restricted to additive or separable forms such as $\sum_e \psi(\mathbf{p}_e)$ and can capture more general dependencies across experts.

**The estimation challenge.** Optimizing (7) directly with stochastic gradient descent is problematic. Since $\bar{\mathbf{p}}(\theta)$ is an expectation over the dataset, it must be estimated, and using the local mean of a mini-batch $\mathcal{B}$, denoted as $\hat{\mathbf{p}} = \frac{1}{|\mathcal{B}|} \sum_{x \in \mathcal{B}} \mathbf{p}(\boldsymbol{x}; \theta)$, introduces significant bias. Because $\phi$ is non-linear, the expectation of the function is not the function of the expectation:

$$\mathbb{E}_{\mathcal{B}}[\phi(\hat{\mathbf{p}})] \neq \phi(\mathbb{E}_{\mathcal{B}}[\hat{\mathbf{p}}]) = \phi(\bar{\mathbf{p}}(\theta)). \tag{8}$$

For small batch sizes, *this bias artificially forces the router to balance every individual mini-batch rather than the global distribution*, potentially degrading performance.

**Duality and mirror descent.** To address the estimation challenges induced by batch-wise statistics, we leverage convex duality to decouple population-level estimation from per-batch updates. Using the identity

$$\phi(\mathbf{p}) = \sup_{\mathbf{q} \in \mathbb{R}^E} \langle \mathbf{p}, \mathbf{q} \rangle - \phi^*(\mathbf{q}), \tag{9}$$

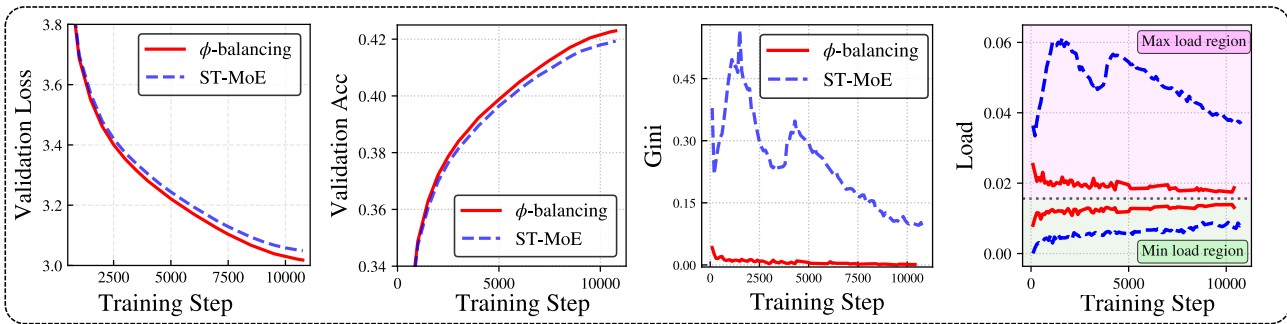

*Figure 3.* **Pre-Training dynamics and expert utilization.** We compare $\phi$-balancing (red, solid) against ST-MoE (blue, dashed) over 10k steps. **(Left) Validation Loss and Accuracy** show that $\phi$-balancing (negative entropy) achieves comparable or superior convergence. **(Right) Gini coefficient and Expert Loading Analysis** demonstrates significantly lower routing imbalance for $\phi$-balancing. $\phi$-balancing maintains tighter bounds between maximum and minimum expert load, staying closer to the perfect allocation line (green) compared to ST-MoE, which exhibits higher variance in expert capacity usage.

we obtain the min-max problem

$$\min_{\theta} \max_{\mathbf{q}\in\mathbb{R}^E} \left( \mathbb{E}_{\boldsymbol{x}\sim\mathcal{D}}[\langle\mathbf{p}(\boldsymbol{x};\theta),\mathbf{q}\rangle] - \phi^*(\mathbf{q}) \right), \quad (10)$$

where $\mathbf{q} \in \mathbb{R}^E$ denotes the dual variable. Intuitively, each component $\mathbf{q}_e$ represents the accumulated congestion cost of expert $e$. When an expert becomes over-utilized (large $\mathbf{p}_e$), its price $\mathbf{q}_e$ increases, amplifying the penalty $\langle\mathbf{p},\mathbf{q}\rangle$ in the primal objective. This encourages the router to shift probability mass toward under-utilized experts.

For any fixed $\theta$, strict convexity and the first-order optimality condition imply that the inner maximization problem admits a unique maximizer given by $\mathbf{q}^{\star} = \nabla\phi(\bar{\mathbf{p}}(\theta))$. Computing $\mathbf{q}^{\star}$ exactly is infeasible in practice, as it requires access to the full data distribution. Moreover, directly applying gradient ascent on the dual variable suffers from high variance when only mini-batch estimates are available. Instead, we exploit the convex structure of the dual problem and adopt mirror descent, which naturally yields a stable online estimator.

Denote by $\mathbf{p}_t$ the empirical mean routing distribution over the mini-batch $\mathcal{B}_t$ at iteration $t$:

$$\mathbf{p}_t := \frac{1}{|\mathcal{B}_t|} \sum_{x\in\mathcal{B}_t} \mathbf{p}(\boldsymbol{x};\theta_t).$$

A single mirror ascent step (Beck & Teboulle, 2003) on the dual objective (10) is equivalent to maintaining an exponential moving average (EMA) of the batch routing distributions followed by a price update:

$$\begin{aligned} \mathbf{m}_{t+1} &\leftarrow (1-\eta)\mathbf{m}_t + \eta\mathbf{p}_t \\ \mathbf{q}_{t+1} &\leftarrow \nabla\phi(\mathbf{m}_{t+1}), \end{aligned} \quad (11)$$

where $\mathbf{m}$ represents the primal variable corresponding to $\mathbf{q}$ and $\eta \in (0, 1]$ is the step size.

The full derivation is provided in Appendix B.

Using $\mathbf{q}_{t+1}$ as an approximation for $\mathbf{q}^{\star}$, the loss w.r.t. $\theta$ becomes

$$\mathcal{L}_{\text{aux}} = \langle\mathbf{p}_t, \mathbf{q}_{t+1}\rangle = \sum_{e=1}^{E} \mathbf{p}_{t,e}\nabla\phi(\mathbf{m}_{t+1})_e, \quad (12)$$

which yields Algorithm 1. Note that we should apply the stop-gradient operator to $\mathbf{m}_{t+1}$ (and hence $\mathbf{q}_{t+1}$) when optimizing the router, so that gradients flow only through $\mathbf{p}_t$.

**Related methods.** As shown in Algorithm 1, our method differs from the ST-MoE loss only in replacing the realized frequency $f_e$ in $\mathcal{L}_{\text{switch}} \propto \sum_e f_e \cdot \mathbf{p}_e$ with our $\nabla\phi(\mathbf{m}_{t+1})_e$. The *hard dispatch fraction* $f_e$ (the percentage of tokens actually sent to expert $e$) introduces discrete, non-differentiable assignment noise. In contrast, our method relies solely on the dual variable $\mathbf{q}$, which tracks the history of the *soft routing probabilities*. Consequently, our regularizer operates entirely within the smooth probability space, avoiding the instability associated with discrete routing decisions.

DeepSeek MoE (Wang et al., 2024; DeepSeek-AI, 2024a) similarly maintains an EMA of recent expert loads to dynamically update per-expert routing-score biases before the top-$k$ decision. However, this approach still relies on the hard routing frequency $f_e$ and does not correspond to principled optimization of a population-level objective as in our derivation.

### 3.2. Examples of $\phi$

The behavior of the min-max LBL (10) is governed by the potential $\phi$. This function determines how the accumulated routing statistics (dual vector $\mathbf{q}$) are mapped to the expert prices (primal vector $\mathbf{m}$) according to (11). We summarize in Table 1 several examples of $\phi$, all of which are strictly convex, symmetric, and differentiable.

*Table 1.* **Summary of $\phi$-balancing variants.** The choice of the potential function $\phi$ determines the relationship between the accumulated expert usage state $\mathbf{m}_{t+1}$ and the auxiliary loss $\mathcal{L}_{\text{aux}}$. Here, summations are taken over the experts $e$, and $q$ denotes the conjugate exponent such that $\frac{1}{p} + \frac{1}{q} = 1$. We follow the convention $0 \log 0 := 0$.

| VARIANT | PRIMAL POTENTIAL $\phi(\mathbf{p})$ | DUAL POTENTIAL $\phi^*(\mathbf{q})$ | AUXILIARY LOSS $\mathcal{L}_{\text{aux}}$ |
|---|---|---|---|
| EUCLIDEAN NORM ($p = 2$) | $\frac{1}{2}\|\mathbf{p}\|_2^2$ | $\frac{1}{2}\|\mathbf{q}\|_2^2$ | $\sum \mathbf{p}_{t,e} \cdot \mathbf{m}_{t+1,e}$ |
| $\ell_p$ NORM ($p > 1$) | $\frac{1}{p}\|\mathbf{p}\|_p^p$ | $\frac{1}{q}\|\mathbf{q}\|_q^q$ | $\sum \mathbf{p}_{t,e} \cdot \text{sgn}(\mathbf{m}_{t+1,e})\|\mathbf{m}_{t+1,e}\|^{p-1}$ |
| SOFT $\ell_1$ NORM ($\delta > 0$) | $\sum(|\mathbf{p}_e| - \delta \log(\frac{1}{\delta}|\mathbf{p}_e| + 1))$ | $\sum -\delta(|\mathbf{q}_e| + \log(1 - |\mathbf{q}_e|))^*$ | $\sum \mathbf{p}_{t,e} \cdot \mathbf{m}_{t+1,e}(|\mathbf{m}_{t+1,e}| + \delta)^{-1}$ |
| NEGATIVE ENTROPY | $\sum \mathbf{p}_e \log \mathbf{p}_e$ | $\sum \exp(\mathbf{q}_e - 1)$ | $\sum \mathbf{p}_{t,e} \cdot (\log \mathbf{m}_{t+1,e} + 1)$ |
| TSALLIS ENTROPY ($\alpha > 0, \alpha \neq 1$) | $\sum(\mathbf{p}_e^\alpha - \mathbf{p}_e)(\alpha - 1)^{-1}$ | no simple closed form | $\sum \mathbf{p}_{t,e} \cdot (\alpha \mathbf{m}_{t+1,e}^{\alpha-1} - 1)(\alpha - 1)^{-1}$ |
| RÉNYI ENTROPY ($\alpha \in (0, 1)$) | $\frac{1}{\alpha - 1} \log(\sum \mathbf{p}_e^\alpha)$ | no simple closed form | $\sum \mathbf{p}_{t,e} \cdot (\alpha \mathbf{m}_{t+1,e}^{\alpha-1})((\alpha - 1)\sum \mathbf{m}_j^\alpha)^{-1}$ |
| PSEUDO-HUBER ($\delta > 0$) | $\sum(\sqrt{\mathbf{p}_e^2 + \delta^2} - \delta)$ | $\sum(-\delta\sqrt{1 - \mathbf{q}_e^2} + \delta)^\dagger$ | $\sum \mathbf{p}_{t,e} \cdot \mathbf{m}_{t+1,e}(\mathbf{m}_{t+1,e}^2 + \delta^2)^{-\frac{1}{2}}$ |
| LOG-COSH ($\beta > 0$) | $\sum \frac{1}{\beta} \log \cosh(\beta \mathbf{p}_e)$ | $\sum(\frac{1+\mathbf{q}_e}{2\beta} \log(1 + \mathbf{q}_e) + \frac{1-\mathbf{q}_e}{2\beta} \log(1 - \mathbf{q}_e))^\ddagger$ | $\sum \mathbf{p}_{t,e} \cdot \tanh(\beta \mathbf{m}_{t+1,e})$ |
| SOFTPLUS | $\sum \log(\exp(\mathbf{p}_e) + 1)$ | $\sum(\mathbf{q}_e \log \mathbf{q}_e + (1 - \mathbf{q}_e) \log(1 - \mathbf{q}_e))^\S$ | $\sum \mathbf{p}_{t,e} \cdot (\exp(-\mathbf{m}_{t+1,e}) + 1)^{-1}$ |

$^*$when $\|\mathbf{q}\|_\infty < 1$, otherwise $\infty$ $\quad^\dagger$when $\|\mathbf{q}\|_\infty \leq 1$, otherwise $\infty$ $\quad^\ddagger$when $|\mathbf{q}_e| < 1$, otherwise 0 when $|\mathbf{q}_e| = 1$, otherwise $\infty$ $\quad^\S$when $\mathbf{q} \in [0, 1]^E$, otherwise $\infty$

**Euclidean potential.** Setting the potential to the squared Euclidean norm $\phi(\mathbf{m}) = \frac{1}{2}\|\mathbf{m}\|_2^2$ yields the conjugate $\phi^*(\mathbf{q}) = \frac{1}{2}\|\mathbf{q}\|_2^2$. Since the link function is defined as $\mathbf{q} = \nabla\phi(\mathbf{m})$, this choice induces the identity map, effectively equating the price vector to the state: $\mathbf{q}_{t+1} = \mathbf{m}_{t+1}$.

**$\ell_p$ potentials.** A simple smooth family parameterized by $p > 1$ is

$$\phi(\mathbf{m}) = \frac{1}{p}\|\mathbf{m}\|_p^p = \frac{1}{p}\sum_{e=1}^E \mathbf{m}_e^p,$$

which yields the link function $\mathbf{q} = \nabla\phi(\mathbf{m}) = \mathbf{m}^{p-1}$ (since $\mathbf{m} \in \Delta^E$ is nonnegative). The exponent $p$ controls the elasticity of the pricing mechanism:

- $p \to 1$ *(dampened):* The exponent vanishes, driving prices toward uniformity regardless of usage history. This effect is also approximated by the *soft $\ell_1$ potential* with

$$\phi(\mathbf{m}) = \|\mathbf{m}\|_1 - \delta \left\|\log\left(\frac{1}{\delta}|\mathbf{m}| + 1\right)\right\|_1,$$

and link function

$$\mathbf{q} = \nabla\phi(\mathbf{m}) = \frac{\mathbf{m}}{|\mathbf{m}| + \delta}.$$

- $p \to \infty$ *(aggressive):* The exponent diverges, causing small disparities in usage to result in extreme price penalties.

**Negative Shannon entropic potential.** Setting $\phi(\mathbf{m}) = \sum \mathbf{m}_e \log \mathbf{m}_e$ yields the dual relationship

$$\mathbf{q} = \nabla\phi(\mathbf{m}) = \log(\mathbf{m}) + \mathbf{1}.$$

This establishes an exponential link between the primal distribution and the dual prices, i.e. $\mathbf{m}_e \approx \exp(\mathbf{q}_e)$. Unlike the linear response, this penalizes low-probability experts aggressively, effectively acting as a soft barrier.

**Negative Tsallis entropic potential.** The negative Tsallis entropy is parameterized by $\alpha > 0$ and $\alpha \neq 1$ and defined as

$$\phi(\mathbf{m}) = \sum_{e=1}^E \frac{\mathbf{m}_e^\alpha - \mathbf{m}_e}{\alpha - 1},$$

with gradient

$$\nabla\phi(\mathbf{m}) = \frac{\alpha \mathbf{m}^{\alpha-1} - 1}{\alpha - 1}.$$

It converges to the negative Shannon entropy in the limit as $\alpha \to 1$.

**Negative Rényi entropic potential.** Another family that generalizes the negative Shannon entropy is the negative Rényi entropy, parameterized by $\alpha \in (0, 1)$ and defined as

$$\phi(\mathbf{m}) = \frac{1}{\alpha - 1} \log\left(\sum_{e=1}^E \mathbf{m}_e^\alpha\right)$$

with gradient

$$\nabla\phi(\mathbf{m}) = \frac{\alpha \mathbf{m}^{\alpha-1}}{(\alpha - 1)\sum_{j=1}^E \mathbf{m}_j^\alpha}.$$

It converges to the negative Shannon entropy in the limit as $\alpha \to 1$.

**Robust potentials.** There are several choices of $\phi$ based on *smooth robust losses*, whose sigmoidal gradients control how aggressively large usage disparities translate into prices. The *pseudo-Huber potential* behaves quadratically in a neighborhood of the origin but smoothly transitions to an approximately linear regime, thereby limiting the influence of extreme outliers. The precise transition scale is controlled by the parameter $\delta > 0$. Similar properties

are enjoyed by the *log-cosh* and *softplus potentials*, whose respective link functions $\mathbf{q} = \tanh(\beta\mathbf{m})$ and

$$\mathbf{q} = \sigma(\mathbf{m}) := \frac{1}{\exp(-\mathbf{m}) + 1}$$

are especially well-behaved.

## 4. Experiments

We evaluate $\phi$-balancing across a range of settings and find that $\phi$-balancing with negative entropy consistently performs best (Figure 3), outperforming Switch-style and loss-free load-balancing baselines across model scales, architectures, and downstream tasks. In large-scale Gemma pretraining, $\phi$-balancing yields more stable routing, lower validation loss, and substantially reduced capacity violations when varying model scale, expert count, and granularity. In downstream fine-tuning, these stability gains translate into stronger task performance and more consistent expert specialization across domains. Our ablations show that history-aware population tracking is critical for robustness, and that entropy-based potentials provide the best overall trade-off between routing stability and downstream accuracy.

### 4.1. Gemma-based Language Model Pretraining

We first evaluate $\phi$-balancing on MoE-augmented Gemma language models (Liang et al., 2025). Unless otherwise stated, all models use top-2 routing and are trained on C4 (Raffel et al., 2020) with the same Gemma-style pretraining recipe (see Appendix C for details on hyperparameters) using negative entropy as $\phi$. Following the settings in Tian et al. (2026), we systematically vary (i) the number of *active* parameters $N$, (ii) the number of experts $E$; and (iii) the MoE *granularity* $G$ (Figure 2).

**Scaling active parameters.** To study how $\phi$-balancing behaves across model scales, we train a family of MoE Transformers with $E = 16$ experts and $A = 2$ active experts per token, and vary the number of active parameters $N$ in $\{111M, 338M, 588M, 986M\}$. Here, $N$ counts only the parameters that are touched for a single token under top-2 routing. For each scale, we compare $\phi$-balancing against standard Switch-style load balancing and loss-free load balancing. We see that the proposed $\phi$-balancing strategy consistently outperforms both baselines across all tested model scales, achieving the lowest validation loss at the 986M parameter mark.

**Scaling the number of experts.** Next, we fix the total active parameter budget and per-token compute, and vary the number of experts $E \in \{8, 16, 32, 64, 128\}$, keeping the number of active experts at $A = 2$. As $E$ increases, we proportionally reduce the size of each expert so that

*Table 2.* **Ablation on mirror maps $\phi$.** We report validation loss and maximum global load-balance violation (MaxVio$_{\text{global}}$), defined in Appendix A; lower is better.

| Family | Mirror map $\phi$ | Val. loss ↓ | MaxVio$_{\text{global}}$ ↓ |
|---|---|---|---|
| **Norm-based potentials** | | | |
| $\ell_p$ norm | $p = 1$ | 3.142 | 0.770 |
| $\ell_p$ norm | $p = 2$ | 3.098 | 0.610 |
| $\ell_p$ norm | $p = 3$ | 3.103 | 0.640 |
| $\ell_p$ norm | $p = \infty$ | 3.116 | 0.760 |
| **Entropic potentials** | | | |
| Entropy | Negative Shannon | **3.084** | **0.104** |
| Entropy | Negative Tsallis ($\alpha = 1.10$) | 3.110 | 0.402 |
| Entropy | Negative Rényi ($\alpha = 0.95$) | 3.091 | 0.207 |
| **Robust potentials** | | | |
| Robust | Soft $\ell_1$ ($\delta > 0$) | 3.109 | 0.740 |
| Robust | Pseudo-Huber ($\delta > 0$) | 3.112 | 0.750 |
| Robust | Log-cosh ($\beta > 0$) | 3.110 | 0.745 |
| Robust | Softplus | 3.125 | 0.810 |

the total FLOPs per token remain approximately constant. This isolates the effect of expert multiplicity, allowing us to test whether $\phi$-balancing continues to stabilize routing when many small experts are available. We see that the performance gap between $\phi$-balancing and both baselines is maintained across the entire range of activation ratios, indicating that the benefit of $\phi$-balancing is robust to the level of model sparsity.

**Scaling granularity.** Finally, we study the effect of MoE granularity by varying the granularity factor $G \in \{2, 4, 8, 16, 32\}$, defined as $G = d_{\text{ff}}/d_{\text{expert}}$, where $d_{\text{expert}}$ denotes the hidden dimension of a single expert and $d_{\text{ff}}$ is the total feed-forward dimension of the MoE layer. Increasing $G$ increases the total number of experts while proportionally decreasing the size of each expert, so that the overall capacity and per-token compute remain fixed and the activation ratio $A/E$ is held constant. Intuitively, larger $G$ corresponds to slicing the feed-forward capacity into finer-grained experts. This setting is particularly sensitive to routing instability, and serves as a stress test for $\phi$-balancing versus conventional load-balancing losses.

**Ablation on mirror maps $\phi$.** As summarized in Table 2, we ablate the choice of mirror map by training identical models with the same $\phi$-balancing objective, router, and hyperparameters, and varying only the potential $\phi$ used to compute the updates (11) with stop-gradient. The potentials are taken from Table 1 in Section 3.2. For each $\phi$, we report validation loss and the global balance metric MaxVio$_{\text{global}}$, computed from the deviation of the held-out global mean routing distribution $\bar{\mathbf{p}}$ from uniform. For the negative Tsallis and Rényi entropies, we sweep several values of $\alpha$ and find that performance is maximized near the Shannon limit $\alpha \to 1$ (Figure 8 in Appendix D).

*Table 3.* **Ablation on global batch size.** Accuracy across methods and global batch sizes on the 986M active-parameter Gemma-MoE with $E = 16$ and $A = 2$. Higher is better. Best per method/column is in **bold**.

| Method | Batch Size | HellaSwag ↑ | MMLU ↑ | C-Eval ↑ |
|--------|-----------|-------------|--------|----------|
| ST-MoE | 32 | 62.82 | 41.96 | 42.58 |
| | 128 | 63.14 | 42.37 | 43.24 |
| | 512 | **63.34** | **42.74** | **43.87** |
| Loss-free | 32 | 62.38 | 41.58 | 42.87 |
| | 128 | 62.73 | 42.03 | 43.46 |
| | 512 | **63.05** | **42.46** | **44.00** |
| Ours | 32 | 63.46 | 42.88 | 43.96 |
| | 128 | 63.60 | 43.02 | 44.18 |
| | 512 | **63.70** | **43.18** | **44.36** |

*Table 4.* **Ablation on EMA tracking choice (routing probabilities vs. selection frequencies).** We compare using an EMA of expert routing probabilities $\mathbf{p}_e$ against using an EMA of selection frequencies $f_e$ as in Algorithm 3.

| $N$ | Dense | Frequency EMA | Probability EMA |
|-----|-------|---------------|-----------------|
| 111M | 3.3768 | 3.089 | **3.0847** |
| 338M | 3.0136 | **2.812** | 2.8200 |
| 588M | 2.8611 | 2.685 | **2.6800** |
| 986M | 2.7142 | **2.598** | 2.6019 |

**Ablation on global batch size.** Table 3 investigates the effect of global batch size on the 986M Gemma-MoE. Unsurprisingly, increasing the batch size improves downstream accuracy across all methods and benchmarks; however, $\phi$-balancing notably outperforms the strongest baselines even at smaller batch sizes, suggesting more effective population-level balancing. Compared with ST-MoE and loss-free balancing, $\phi$-balancing delivers both higher absolute accuracy and greater robustness to global batch size, making it the most effective choice in this ablation.

**Ablation on EMA load tracking.** Table 4 studies how the choice of load statistic affects training. By default, we track expert load using an EMA of the router's *pre-top-k* assignment probabilities, which provides a smooth estimate of expected utilization. As an alternative, we replace this with an EMA of realized selection frequencies ($f_e$). As shown in Table 4, using the EMA of frequencies yields performance comparable to using the EMA of probabilities.

**EMA decay sensitivity.** We study how the EMA decay $\eta$ used to maintain the running estimate of global routing statistics affects training dynamics. Keeping the model, router, loss weights, and optimization settings fixed, we sweep $\eta$ over $[0, 1]$, and rerun training under identical conditions. For each setting, we evaluate validation loss and accuracy at convergence, and plot both metrics against $\eta$ in Figure 4.

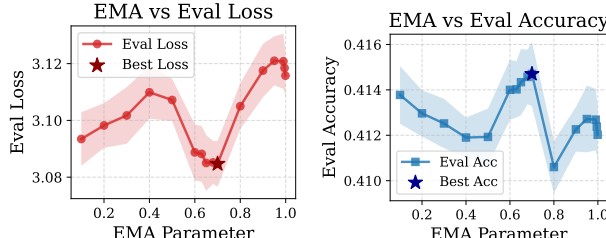

*Figure 4.* **Sensitivity analysis of the EMA decay parameter $\eta$.** Validation loss (red; left) and accuracy (blue; right) are shown as $\eta$ varies over $[0, 1]$.

We see that the best trade-off is achieved for $\eta \in [0.6, 0.7]$. Performance becomes unstable at high decay, where load estimates revert to single-batch statistics and exhibit high variance.

### 4.2. Downstream Fine-Tuning

We now evaluate $\phi$-balancing on three large MoE backbones with distinct architectures: **DeepSeek-MoE-16B-Chat** (Dai et al., 2024), **DeepSeek-V2-Lite-Chat** (DeepSeek-AI, 2024a), and **Moonlight-16B-A3B-Instruct** (Liu et al., 2025). We report results on seven benchmarks. For training, we use the training sets from Numina (Li et al., 2024), and below benchmarks which cover three categories: (i) Mathematics—GSM8K (Cobbe et al., 2021) and MATH500 (Lightman et al., 2024); (ii) Multi-domain tasks—BBH (Suzgun et al., 2023), GLUE training mixture (Socher et al., 2013; Rajpurkar et al., 2016; Williams et al., 2018; Warstadt et al., 2019; Wang et al., 2019), LiveBench (Zhang et al., 2025c), and GPQA (Rein et al., 2024); and (iii) Code generation—HumanEval (Chen et al., 2021). The results are shown in Table 5 and Figure 5.

**Per-benchmark fine-tuning.** To isolate specialization behavior and avoid confounding from heterogeneous multi-task mixing, we fine-tune *each benchmark separately*. For each benchmark, we construct a 6,000-example training set by uniformly sampling 6,000 prompts from the benchmark's training distribution when available; benchmarks with fewer than 6,000 total examples are supplemented to 6,000 with NuminaTest examples (Li et al., 2024). All training targets include high-quality chain-of-thought reasoning produced by a strong teacher model (OpenAI GPT-5.2), ensuring consistent supervision quality across tasks and models and complementing recent advances in reasoning-focused post-training, e.g. guided adversarial self-play (Li et al., 2026).

**Optimization and adapters.** All single benchmark runs use AdamW (Su et al., 2026) with learning rate $5 \times 10^{-5}$, weight decay 0.01, and 500 warmup steps. We perform parameter-efficient fine-tuning via LoRA with rank $r = 8$, $\alpha = 32$, dropout 0.1, and apply adapters to q_proj, k_proj, v_proj, o_proj, gate_proj, up_proj, and

*Table 5.* **Domain specialization.** Accuracy across benchmarks (mean $\pm$ approx. std (half-width/1.96); higher is better). **Avg** is the unweighted mean over all benchmarks shown. Best per model/column is in **bold**.

| Model | Method | Multi-Domain | | | | Code | Math | | Avg |
| | | BBH | GLUE | LiveBench | GPQA | HumanEval | GSM8K | Math500 | Avg |
|---|---|---|---|---|---|---|---|---|---|
| DeepSeek-MoE-Chat | Frozen checkpoint | $33.07 \pm 2.08$ | $55.97 \pm 0.64$ | $5.92 \pm 0.62$ | $28.91 \pm 1.28$ | $39.63 \pm 3.78$ | $57.63 \pm 0.93$ | $14.80 \pm 1.59$ | 33.70 |
| | ST-MoE | $69.86 \pm 2.03$ | $79.03 \pm 0.53$ | $14.62 \pm 0.93$ | $79.70 \pm 1.14$ | **$41.46 \pm 3.81$** | $64.00 \pm 0.91$ | $15.00 \pm 1.60$ | 51.95 |
| | **Ours** | **$73.92 \pm 1.80$** | **$80.41 \pm 0.50$** | **$17.85 \pm 0.87$** | **$82.34 \pm 1.03$** | $40.21 \pm 3.88$ | **$66.28 \pm 0.83$** | **$16.40 \pm 1.50$** | **53.92** |
| DeepSeek-V2-Lite | Frozen checkpoint | $35.42 \pm 2.11$ | $53.12 \pm 0.64$ | $13.93 \pm 0.91$ | $19.98 \pm 1.13$ | $37.20 \pm 3.73$ | **$69.20 \pm 0.87$** | $19.40 \pm 1.77$ | 35.46 |
| | ST-MoE | $57.34 \pm 2.18$ | **$74.88 \pm 0.56$** | $16.85 \pm 0.99$ | $68.67 \pm 1.31$ | $45.12 \pm 3.84$ | $62.00 \pm 0.92$ | $20.20 \pm 1.79$ | 49.29 |
| | **Ours** | **$61.98 \pm 1.98$** | $74.35 \pm 0.59$ | **$19.42 \pm 0.95$** | **$72.91 \pm 1.18$** | **$48.36 \pm 3.62$** | $64.38 \pm 0.85$ | **$21.60 \pm 1.64$** | **51.86** |
| Moonlight-16B-A3B -Instruct | Frozen checkpoint | $59.10 \pm 2.17$ | $60.68 \pm 0.63$ | $19.57 \pm 1.05$ | $27.98 \pm 1.27$ | $72.56 \pm 3.45$ | $92.84 \pm 0.49$ | **$67.40 \pm 2.09$** | 57.16 |
| | ST-MoE | $82.00 \pm 1.70$ | $81.48 \pm 0.50$ | $42.21 \pm 1.35$ | $78.59 \pm 1.16$ | $73.17 \pm 3.43$ | $91.37 \pm 0.53$ | $64.80 \pm 2.13$ | 73.37 |
| | **Ours** | **$85.74 \pm 1.54$** | **$83.02 \pm 0.46$** | **$46.83 \pm 1.24$** | **$81.92 \pm 1.05$** | **$75.88 \pm 3.24$** | **$92.92 \pm 0.57$** | $66.20 \pm 2.02$ | **76.07** |

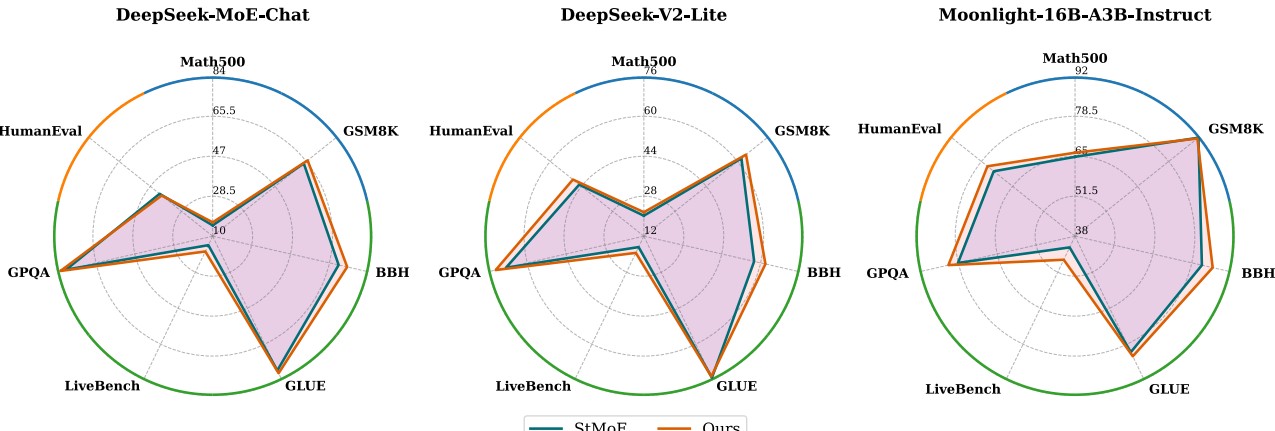

*Figure 5.* **Performance comparison of ablation method combinations across three model architectures.** The radar charts illustrate the evaluation of DeepSeek-MoE-Chat, DeepSeek-V2-Lite, and Moonlight-16B-A3B-Instruct on seven diverse benchmarks. Radial axes represent the corresponding benchmark scores, identified by the labels and the color-coded outer rim segments. The proposed method (*Ours*) demonstrates a consistent expansion of the performance capabilities compared to the ST-MoE baseline.

`down_proj`. We train for 3 epochs with global batch size 18 (about 1,000 optimization steps).

**Tuning protocol.** To ensure a fair comparison across routing regularizers, we independently tune—*for each model and each method*—the learning rate, the load-balancing loss coefficient, and the history-regularization weight $\eta$ using a held-out validation split drawn from the training pool, while keeping all other hyperparameters fixed. This yields an extensive experimental grid spanning 3 backbones $\times$ 7 benchmarks $\times$ multiple routing regularizers.

**Observations.** As shown in Table 5, $\phi$-balancing achieves state-of-the-art results in over 80% of the 21 setups across three models. In some setups, $\phi$-balancing even outperforms the next-best method by nearly 5%. We see that the average performance across the benchmarks is consistently higher for $\phi$-balancing. We also observe $\phi$-balancing's performance is better than ST-MoE's over 90% of the time.

**Domain specialization.** The robust and history-aware regularization of $\phi$-balancing promotes the emergence of diverse expert behaviors. By balancing expert usage at the population level rather than forcing uniformity within each mini-batch, $\phi$-balancing enables experts to specialize along different functional dimensions, allowing the router to combine complementary experts in a manner reminiscent of ensemble methods (Figure 6). This behavior contrasts with ST-MoE, whose batch-level load-balancing objective encourages the tokens in each mini-batch to be distributed uniformly across experts, effectively pushing every expert to learn every domain and suppressing specialization.

## 5. Related Work

**Foundational MoE architectures and theory.** MoE routing has evolved from early load-balancing heuristics (Shazeer et al., 2017) to scalable systems such as GShard (Lepikhin et al., 2021) and Switch Transformers (Fedus et al., 2022). This architecture has since become the backbone of modern frontier models, including DeepSeek-V3 (DeepSeek-AI, 2024b), Qwen3 (Qwen Team, 2025), and OLMoE (Muennighoff et al., 2025). Subsequent research has explored diverse variants such as Expert Choice Routing

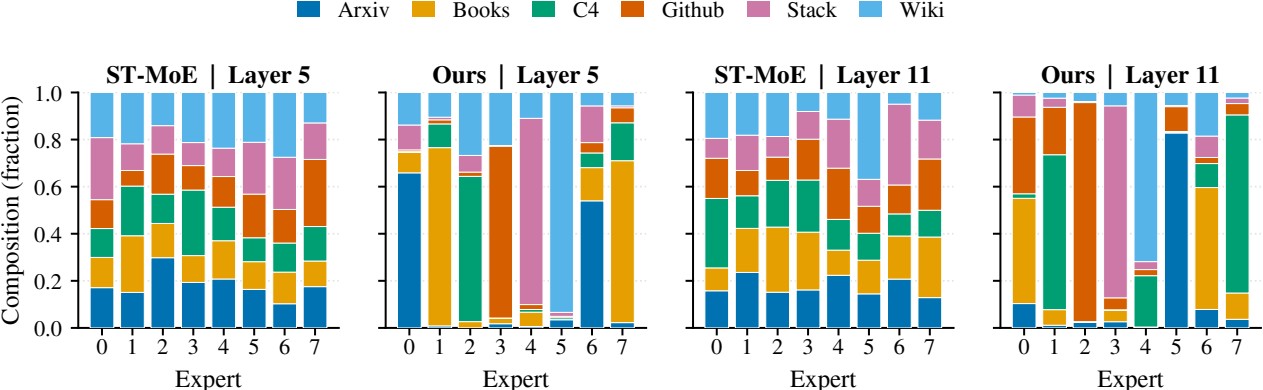

*Figure 6.* **Domain specialization in routing.** Routed-token ratio (fraction of tokens) assigned to each expert (IDs 0–7) for different data domains (Arxiv, Books, C4, Github, Stack, Wiki) at two representative layers (Layer 5 and Layer 11). Compared to ST-MoE, our router exhibits sharper domain-to-expert preferences (stronger specialization), albeit with mildly uneven expert loads.

(Zhou et al., 2022), DeepSeekMoE's fine-grained segmentation (Dai et al., 2024), scaling laws for efficient dispatch (Tian et al., 2026), and applications to different settings (Xu et al., 2026; Zhang et al., 2025a). Concurrently, theoretical understanding has deepened: recent works have established convergence guarantees for gating mechanisms (Nguyen et al., 2024; Le et al., 2026; Thai et al., 2025; Nguyen et al., 2026) and analyzed MoE optimization dynamics through the lens of mirror descent (Fruytier et al., 2025). We refer readers to comprehensive surveys on MoE models (Cai et al., 2025; Mu & Lin, 2025; Zhang et al., 2025b).

**Refining load balancing.** The standard auxiliary load-balancing loss has faced scrutiny for suppressing expert specialization (Qiu et al., 2025b; Guo et al., 2025) and introducing gradient interference (Qiu et al., 2025a). Proposed remedies range from calculating losses over global batches (Qiu et al., 2025b) to exploiting orthogonality (Omi et al., 2025; Cheng et al., 2025) and utilizing ternary rewards (Yan et al., 2025). Alternatively, some methods modify the LBL (Huang et al., 2024; Zeng et al., 2024; Lv et al., 2026), remove auxiliary gradients entirely (Wang et al., 2024; Yang, 2025), or propose fully differentiable routing mechanisms like ReMoE (Wang et al., 2025b) to bypass discrete selection issues. While Dai et al. (2022) address the resulting routing instability via two-stage distillation, these approaches largely rely on heuristics or specific structural constraints.

**Connections to optimization.** The core mechanism of $\phi$-balancing is based on mirror descent, which has roots in the classical optimization literature (Nemirovski & Yudin, 1983; Beck & Teboulle, 2003) as a generalization of gradient descent to the geometry induced by a chosen, strongly convex mirror map. More broadly, the use of convex potential functions as an algorithmic design primitive has also been explored in the generalization of Lion (Chen et al., 2023;

Liu et al., 2024; Chen, 2025) to its Lion-$\mathcal{K}$ variants (Chen et al., 2024; 2026b), in which sign-based update rules are replaced by subgradients of more general convex functions.

## 6. Conclusion

We introduced $\phi$-balancing, a simple and theoretically principled framework for balancing expert utilization in MoE models. At its core, $\phi$-balancing leverages a strictly convex, symmetric, and differentiable potential to encourage population-level routing probabilities toward uniformity, yielding an auxiliary objective whose online mirror descent updates are equivalent to maintaining an EMA of expert loads. Empirically, $\phi$-balancing consistently improves over standard baselines across a wide range of pretraining and fine-tuning tasks and benchmarks. The entire mechanism incurs negligible overhead and requires minimal changes to existing MoE pipelines, making it suitable for large-scale deployments.

**Future work.** The breadth of the $\phi$-balancing framework opens several promising directions. A particularly important one is to develop a clearer understanding of the role of $\phi$, including which choices are optimal in different regimes, why certain choices induce more uniform routing dynamics, and how these choices scale. Beyond this, $\phi$-balancing provides a natural foundation for richer MoE settings, including expert parallelism with heterogeneous capacities (Wang et al., 2025a), multimodal data distributions (Zhao et al., 2020), and curriculum-based schedules for routing noise. Finally, we expect $\phi$-balancing to lead to compounding gains when combined with recent advances in optimization, including online subspace descent (Liang et al., 2024), Muon (Jordan et al., 2024; Liu et al., 2025), H-Fac (Nguyen et al., 2025), DeMo (Peng et al., 2026), and cautious variants (Liang et al., 2026; Chen et al., 2026a).

## Acknowledgments

We thank the anonymous reviewers, area chairs, and senior area chairs for their careful evaluation and constructive feedback, as well as Ziteng Wang and Hongcan Guo for helpful discussions.

## Impact Statement

This paper presents work whose goal is to advance the field of machine learning. There are many potential societal consequences of our work, none of which we feel must be specifically highlighted here.

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

---

**Algorithm 2** Detailed version of Algorithm 1

---

**Require:** task learning rate $\gamma$, strictly convex, symmetric, and differentiable function $\phi$, momentum $\eta \in (0, 1]$, $\phi$-balancing weight $\alpha > 0$, history vectors $\mathbf{m}^{(l)} \leftarrow \mathbf{0}$ for each MoE layer $l = 1, \ldots, L$, model parameters $\theta$

1: **while** training **do**
2:      Sample mini-batch $\mathcal{B}$ from dataset.
3:      $\mathcal{L}_{\text{task}}, \{\mathbf{p}^{(l)}\}_{l=1}^{L} \leftarrow \text{FORWARD}(\theta, \mathcal{B})$               (compute task loss and mean routing probabilities)
4:      $\mathcal{L}_{\text{total}} \leftarrow \mathcal{L}_{\text{task}}$
5:      **for** each MoE layer $l = 1$ to $L$ **do**
6:          $\mathbf{m}^{(l)} \leftarrow (1 - \eta)\mathbf{m}^{(l)} + \eta\mathbf{p}^{(l)}$                            (EMA of loads)
7:          $\mathcal{L}_{\text{aux}}^{(l)} \leftarrow \sum_{e=1}^{E} \mathbf{p}_e^{(l)} \cdot \text{STOPGRAD}(\nabla\phi(\mathbf{m}^{(l)})_e)$
8:          $\mathcal{L}_{\text{total}} \leftarrow \mathcal{L}_{\text{total}} + \alpha \cdot E \cdot \mathcal{L}_{\text{aux}}^{(l)}$
9:      **end for**
10:     $\theta \leftarrow \text{OPTIMIZER}(\theta; \mathcal{L}_{\text{total}})$
11: **end while**

---

**Algorithm 3** $\phi$-balancing with Frequency EMA for one MoE layer

---

**Require:** strictly convex, symmetric, and differentiable function $\phi$, $\eta \in (0, 1]$, $\alpha > 0$, $\mathbf{m} \leftarrow \mathbf{0}$, routing frequencies $f_e$ defined in (4)

1: Compute routing probabilities $p_{i,e}$ for each token $i$
2: $\mathbf{p}_e \leftarrow \frac{1}{T}\sum_{i=1}^{T} p_{i,e}$ for $e = 1, \ldots, E$                               (expert loads)
3: $\mathbf{p} \leftarrow (\mathbf{p}_1, \ldots, \mathbf{p}_E)$
4: $\mathbf{f}_e \leftarrow \frac{1}{T}\sum_{i=1}^{T} \mathbb{I}(e \in \text{Top-}k(p_{i,.}))$ for $e = 1, \ldots, E$           (selection frequencies)
5: $\mathbf{f} \leftarrow (\mathbf{f}_1, \ldots, \mathbf{f}_E)$
6: $\mathbf{m} \leftarrow (1 - \eta)\mathbf{m} + \eta\mathbf{f}$                                    (EMA of frequencies)
7: $\mathcal{L}_{\text{aux}} \leftarrow \sum_{e=1}^{E} \nabla\phi(\mathbf{m})_e \, \mathbf{p}_e$
8: Update model using $\nabla(\mathcal{L}_{\text{task}} + \alpha \cdot E \cdot \mathcal{L}_{\text{aux}})$

---

# Appendix

## A. Notation

$\mathbf{0}$ and $\mathbf{1}$ denote the all-zeros and all-ones vectors of the appropriate dimension, respectively. $\|\cdot\|_p$ denotes the $\ell_p$ norm for $p \in [1, \infty]$. $\langle \cdot, \cdot \rangle$ denotes the standard inner product. $\mathbb{I}(\cdot)$ denotes the 0-1 indicator function. $\overline{\mathbb{R}}$ denotes the extended real number line.

$$\Delta^E := \left\{ \mathbf{p} \in \mathbb{R}_{\geq 0}^E : \sum_{e=1}^{E} \mathbf{p}_e = 1 \right\}$$

denotes the probability simplex. The convex conjugate $\phi^*$ of a function $\phi : \mathbb{R}^d \to \overline{\mathbb{R}}$ is defined as

$$\phi^*(\mathbf{y}) := \sup_{\mathbf{x} \in \mathbb{R}^d} \langle \mathbf{x}, \mathbf{y} \rangle - \phi(\mathbf{x}), \quad \text{for all } \mathbf{y} \in \mathbb{R}^d.$$

**MaxVio$_{\text{global}}$** (Wang et al., 2024) is a metric that quantifies load imbalance in MoE models, defined as

$$\text{MaxVio}_{\text{global}} = \frac{\max_e \text{Load}_e - \overline{\text{Load}}}{\overline{\text{Load}}},$$

where

- $\text{Load}_e$ is the number of tokens assigned to expert $e$.
- $\overline{\text{Load}}$ is the average (ideal balanced) load across experts.

A lower value indicates more balanced expert utilization, while a higher value reflects severe imbalance. It evaluates global load balance across the entire validation set, reflecting long-term efficiency and fairness in expert usage.

**Accuracy (ACC)** is a metric that measures the proportion of correct predictions made by a model. It is calculated as the number of correct predictions divided by the total number of predictions:

$$\text{ACC} = \frac{\text{Number of Correct Predictions}}{\text{Total Number of Predictions}}.$$

**Routed-token ratio** is a metric that quantifies expert specialization. Let $\mathcal{T}_d$ denote the set of tokens belonging to domain $d$, and let $g_l(i) \in \{0, \ldots, E-1\}$ be the index of the expert selected by the router for token $i$ at layer $l$.

The routing distribution ratio $R_{e,d}^{(l)}$ for expert $e$, domain $d$, and layer $l$ is calculated as the normalized frequency of token assignment:

$$R_{e,d}^{(l)} = \frac{\sum_{i \in \mathcal{T}_d} \mathbb{I}(g_l(i) = e)}{|\mathcal{T}_d|}.$$

A value of $R_{e,d}^{(l)} \approx \frac{1}{E}$ indicates a uniform, domain-agnostic distribution, whereas $R_{e,d}^{(l)} \gg \frac{1}{E}$ suggests strong domain specialization for expert $e$.

## B. Proofs

**Lemma 1** (Uniform distribution uniquely minimizes strictly convex and symmetric functions). *Let $\phi : \Delta^E \to \mathbb{R}$ be symmetric, i.e.*

$$\phi(\mathbf{P}\mathbf{p}) = \phi(\mathbf{p}) \qquad \text{for all permutation matrices } \mathbf{P} \in \{0,1\}^{E \times E},$$

*and strictly convex. Let $\mathbf{u} := \frac{1}{E}\mathbf{1} \in \Delta^E$ denote the uniform distribution. Then*

$$\phi(\mathbf{u}) \leq \phi(\mathbf{p}) \qquad \text{for all } \mathbf{p} \in \Delta^E,$$

*with equality if and only if $\mathbf{p} = \mathbf{u}$.*

*Proof.* Let $\mathcal{S}_E$ denote the set of all $E!$ permutation matrices on $\mathbb{R}^{E \times E}$, and define the centroid

$$\bar{\mathbf{p}} := \frac{1}{E!} \sum_{\mathbf{P} \in \mathcal{S}_E} \mathbf{P}\mathbf{p}.$$

Since $\Delta^E$ is convex and each $\mathbf{P}\mathbf{p} \in \Delta^E$, we have $\bar{\mathbf{p}} \in \Delta^E$.

Fix any coordinate $j \in \{1, \ldots, E\}$. Across all permutations, each entry $\mathbf{p}_e$ appears in position $j$ exactly $(E-1)!$ times, since the remaining $E-1$ coordinates can be permuted arbitrarily. Hence

$$\bar{\mathbf{p}}_j = \frac{1}{E!} \sum_{\mathbf{P} \in \mathcal{S}_E} (\mathbf{P}\mathbf{p})_j = \frac{(E-1)!}{E!} \sum_{e=1}^{E} \mathbf{p}_e = \frac{1}{E} \cdot 1 = \frac{1}{E},$$

where the penultimate equality uses $\mathbf{p} \in \Delta^E$. Since $j$ is arbitrary, it follows that $\bar{\mathbf{p}} = \mathbf{u}$.

Applying Jensen's inequality to the convex function $\phi$ with the uniform weights $\frac{1}{E!}$, which are nonnegative and sum to 1,

$$\phi(\mathbf{u}) = \phi(\bar{\mathbf{p}}) = \phi\left(\frac{1}{E!} \sum_{\mathbf{P} \in \mathcal{S}_E} \mathbf{P}\mathbf{p}\right) \leq \frac{1}{E!} \sum_{\mathbf{P} \in \mathcal{S}_E} \phi(\mathbf{P}\mathbf{p}) = \phi(\mathbf{p}),$$

where the last equality uses the symmetry of $\phi$. Thus $\phi(\mathbf{u}) \leq \phi(\mathbf{p})$.

Since $\phi$ is strictly convex, Jensen's inequality is strict unless all points in the average are identical, i.e. $\mathbf{P}\mathbf{p} = \mathbf{p}$ for all $\mathbf{P} \in \mathcal{S}_E$. This holds if and only if $\mathbf{p}$ has all coordinates equal, from which we conclude $\mathbf{p} = \mathbf{u}$. $\square$

**Lemma 2** (Mirror ascent step for the inner maximization). *Let $\phi : \mathcal{D}_\phi \to \overline{\mathbb{R}}$ be of Legendre type (proper, strictly convex, and essentially smooth) on an open convex domain $\mathcal{D}_\phi \subseteq \mathbb{R}^d$. Then $\phi^*$ is of Legendre type on $\mathcal{D}_{\phi^*} := \nabla\phi(\mathcal{D}_\phi)$, and the gradient maps $\nabla\phi : \mathcal{D}_\phi \to \mathcal{D}_{\phi^*}$ and $\nabla\phi^* : \mathcal{D}_{\phi^*} \to \mathcal{D}_\phi$ are inverse functions, i.e.*

$$\mathbf{m} = \nabla\phi^*(\mathbf{q}) \iff \mathbf{q} = \nabla\phi(\mathbf{m}) \qquad \textit{for all } \mathbf{m} \in \mathcal{D}_\phi \textit{ and } \mathbf{q} \in \mathcal{D}_{\phi^*}.$$

*Fix $\mathbf{p}_t \in \mathcal{D}_\phi$ and consider the inner maximization objective of* (10)

$$F(\mathbf{q}; \mathbf{p}_t) := \langle \mathbf{p}_t, \mathbf{q} \rangle - \phi^*(\mathbf{q}).$$

*Let $\mathbf{q}_t \in \mathcal{D}_{\phi^*}$ be the current iterate, with corresponding primal representation $\mathbf{m}_t := \nabla\phi^*(\mathbf{q}_t) \in \mathcal{D}_\phi$. Then, for any step size $\eta \in (0, 1]$, the mirror ascent update on $F$ with mirror map $\phi^*$,*

$$\mathbf{q}_{t+1} \leftarrow \underset{\mathbf{q} \in \mathcal{D}_{\phi^*}}{\arg\max} \left\{ \langle \nabla_\mathbf{q} F(\mathbf{q}_t; \mathbf{p}_t), \mathbf{q} \rangle - \frac{1}{\eta} D_{\phi^*}(\mathbf{q}, \mathbf{q}_t) \right\},$$

*where $D_{\phi^*}(\mathbf{p}, \mathbf{q}) := \phi^*(\mathbf{p}) - \phi^*(\mathbf{q}) - \langle \nabla\phi^*(\mathbf{q}), \mathbf{p} - \mathbf{q} \rangle$ is the Bregman divergence induced by $\phi^*$, admits the closed form*

$$\mathbf{m}_{t+1} \leftarrow (1 - \eta)\mathbf{m}_t + \eta\mathbf{p}_t, \qquad \mathbf{q}_{t+1} \leftarrow \nabla\phi(\mathbf{m}_{t+1}).$$

*Proof.* The first assertion is a classical result in convex analysis (Rockafellar, 1970, Theorem 26.5). Differentiating $F$ with respect to $\mathbf{q}$ at the current iterate $\mathbf{q}_t$ yields

$$\nabla_\mathbf{q} F(\mathbf{q}_t; \mathbf{p}_t) = \mathbf{p}_t - \nabla\phi^*(\mathbf{q}_t) = \mathbf{p}_t - \mathbf{m}_t. \tag{13}$$

The first-order optimality condition for the mirror ascent update (Beck & Teboulle, 2003) requires that the gradient of its objective in $\mathbf{q}$ vanish at $\mathbf{q}_{t+1}$, i.e.

$$\nabla_\mathbf{q} F(\mathbf{q}_t; \mathbf{p}_t) - \frac{1}{\eta}(\nabla\phi^*(\mathbf{q}_{t+1}) - \nabla\phi^*(\mathbf{q}_t)) = \mathbf{0}. \tag{14}$$

Substituting (13) and $\nabla\phi^*(\mathbf{q}_t) = \mathbf{m}_t$ into (14) and rearranging yields the primal-space update

$$\mathbf{m}_{t+1} = \nabla\phi^*(\mathbf{q}_{t+1}) \leftarrow \mathbf{m}_t + \eta(\mathbf{p}_t - \mathbf{m}_t) = (1 - \eta)\mathbf{m}_t + \eta\mathbf{p}_t,$$

which is a convex combination of $\mathbf{m}_t$ and $\mathbf{p}_t$. Since $\mathcal{D}_\phi$ is convex, the iterate $\mathbf{m}_{t+1}$ remains in $\mathcal{D}_\phi$. Mapping back to the dual space gives

$$\mathbf{q}_{t+1} \leftarrow \nabla\phi(\mathbf{m}_{t+1}),$$

which completes the proof. $\square$

**Remark 1.** *If $\phi : \mathbb{R}^d \to \mathbb{R}$ is strictly convex and differentiable, then it is of Legendre type on $\mathbb{R}^d$.*

## C. Hyperparameters

We set the tokens-per-parameter budget for all MoE pretraining runs using the MoE scaling law of Tian et al. (2026). For a given total training compute $C$, the compute-per-token $M_{\text{MoE}}^{\text{opt}}$ and the optimal number of training tokens $D_{\text{MoE}}^{\text{opt}}$ are given by

$$M_{\text{MoE}}^{\text{opt}} = 0.1915\, C^{0.5095}, \qquad D_{\text{MoE}}^{\text{opt}} = 5.2232\, C^{0.4905}.$$

The corresponding optimal tokens-per-parameter ratio is

$$\text{tpp}^{\text{opt}}(C) = \frac{D_{\text{MoE}}^{\text{opt}}}{M_{\text{MoE}}^{\text{opt}}} = \frac{5.2232}{0.1915} C^{0.4905 - 0.5095} \approx 27.3\, C^{-0.019},$$

and we choose the total number of training tokens for each configuration to match $\text{tpp}^{\text{opt}}(C)$ induced by our target compute $C$.

*Table 6.* **Model hyperparameters.** Comparison of model configurations across different sizes, ordered by parameter count.

| Model | Architecture | | | | | | Training | | | Optimization | | |
|---|---|---|---|---|---|---|---|---|---|---|---|---|
| | $d_{\text{model}}$ | $L$ | $H$ | $d_{\text{head}}$ | FFN | Act. | Seq. Len | Batch | Total Steps | Peak LR | Warmup | WD |
| **111M** | 512 | 8 | 8 | 64 | 2048 | gelu | 2048 | 256 | 10,758 | 1.65e-3 | 10% | 0.261 |
| **338M** | 1024 | 8 | 8 | 128 | 4096 | gelu | 2048 | 256 | 38,658 | 1.37e-3 | 10% | 0.276 |
| **588M** | 1280 | 10 | 10 | 128 | 5120 | gelu | 2048 | 256 | 20,500 | 1.2e-3 | 10% | 0.29 |
| **986M** | 1536 | 12 | 8 | 256 | 6144 | gelu | 2048 | 512 | 60,751 | 1.0e-3 | 10% | 0.3 |

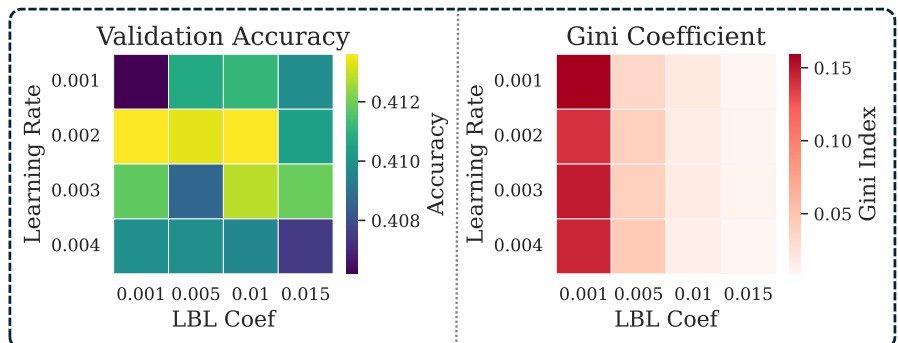

*Figure 7.* **Hyperparameter sensitivity analysis.** Heatmaps displaying **Validation Accuracy** (left) and **Gini Coefficient** (right) across varying Learning Rates ($\gamma \in \{1e\text{-}3, \dots, 4e\text{-}3\}$) and $\phi$-balancing loss coefficient ($\alpha \in \{0.001, \dots, 0.015\}$). While accuracy remains robust (peak 0.4136), increasing $\alpha$ drastically reduces the Gini coefficient.

For the hyperparameters listed in Table 6, our learning rate search employed a quasi-logarithmic grid spanning to $1 \times 10^{-5}$ to $1 \times 10^{-1}$, with denser sampling in the $10^{-4}$ to $10^{-2}$ range where transformer models typically achieve optimal performance. The grid included standard decade values (e.g., 0.001, 0.01) as well as intermediate points within each logarithmic interval (e.g., 0.2, 0.3, 0.5, 0.8 scaled to each decade), totaling 24 distinct learning rate values. For the learning rate schedule, we systematically evaluated warmup ratios of 0, 0.05, 0.1, 0.2, 0.3, 0.4, 0.5 followed by cosine annealing decay.

## D. Additional Experiments

Algorithm 2 details the complete implementation of the routing mechanism introduced in Algorithm 1. A key distinction in this detailed formulation is the load tracking strategy: we specifically utilize the EMA of expert assignment probabilities, whereas alternative approaches typically track the EMA of selection frequencies ($f_e$).

To systematically compare our $\phi$-balancing, ST-MoE, and loss-free MoE models, we categorize our experimental configurations into three distinct scaling regimes as summarized in Table 7. The **Active-Parameter** study follows standard scaling principles by varying the model capacity from 111M to 986M active parameters while maintaining a fixed routing sparsity of 2-of-16. In the **Granularity** study, we investigate the trade-off between expert specialization and parameter count by varying the factor $G$; specifically, we increase the total number of experts $E$ while proportionally shrinking the hidden dimension of each expert's feed-forward network to ensure per-token FLOPs remain invariant. Finally, the **Expert-Count** study isolates the impact of the activation ratio ($A/E$) by holding the individual expert size and compute budget constant while expanding the total expert pool $E$ from 8 to 128. This experimental design allows us to disentangle the benefits of total model capacity from those of routing density and expert specialization.

*Table 7.* **Summary of MoE scaling study configurations.** All studies are conducted while keeping per-token FLOPs approximately constant.

| Scaling Axis | Variable ($x$) | Fixed Constraints | Configurations / Values |
|---|---|---|---|
| **Active-Parameter** | Active Params ($N$) | $E = 16$ 
 $A = 2$ | 111M, 338M, 
 588M, 986M |
| **Granularity** | Factor ($G$) | Model Size ($M$) 
 Ratio ($A/E$) | $G \in \{2, 4, 8, 16, 32\}$ 
 ($E$ scales $16 \to 256$) |
| **Expert-Count** | Total Experts ($E$) 
 (Sparsity) | Compute ($M$), $A = 2$ 
 Expert Size | $E \in \{8, 16, 32, 64, 128\}$ |

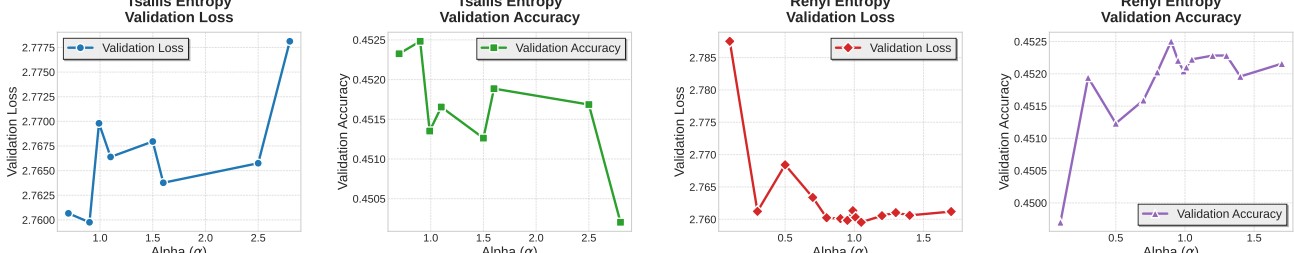

*Figure 8.* **Effect of entropy order $\alpha$ on validation performance.** Validation loss and accuracy are reported after 20.5K training steps for Tsallis and Rényi entropy variants across different values of $\alpha$. Both objectives exhibit a stable accuracy profile near $\alpha \approx 0.9$–$1.0$. Conversely, extreme values of $\alpha$ degrade validation loss and/or accuracy, indicating that moderate entropy orders yield the most robust performance. *Model configuration:* $\sim$588M active parameters per token, with a total memory footprint of $\sim$3.34B parameters.

*Table 8.* **Mixed benchmark.** We combine 1,500 examples from the seven benchmarks used in per-benchmark finetuning, and combine them into a mixed finetuning dataset. Similar to per-benchmark finetuning, each example contains high quality chain-of-thought reasoning from a strong teacher model (OpenAI GPT-5.2). For benchmarks with less than total 1,500 examples, we select all of its training distribution. We finetune with LoRA rank $r = 4$ on one epoch with learning rate 2e-5 (approximately 500 steps). We finetune Deepseek-MoE-16B-Chat and Deepseek-V2-Lite-Chat models and show the accuracy of each benchmark in the evaluation set.

| Model | Method | Multi-Domain | | | | Code | Math | | Avg |
|---|---|---|---|---|---|---|---|---|---|
| | | BBH | GLUE | LiveBench | GPQA | HumanEval | GSM8K | Math500 | Avg |
| DeepSeek-MoE -Chat | Frozen checkpoint | $33.07 \pm 2.08$ | $55.97 \pm 0.64$ | $5.92 \pm 0.62$ | $\mathbf{28.91 \pm 1.28}$ | $\mathbf{39.63 \pm 3.78}$ | $\mathbf{57.63 \pm 0.93}$ | $14.80 \pm 1.59$ | $33.70$ |
| | ST-MoE | $43.05 \pm 2.19$ | $67.72 \pm 0.60$ | $13.79 \pm 0.91$ | $28.75 \pm 1.28$ | $29.27 \pm 3.55$ | $53.33 \pm 0.94$ | $\mathbf{15.54 \pm 1.93}$ | $35.92$ |
| | **Ours** | $\mathbf{44.22 \pm 2.10}$ | $\mathbf{68.51 \pm 0.56}$ | $\mathbf{14.15 \pm 0.87}$ | $27.88 \pm 1.26$ | $28.05 \pm 3.51$ | $54.91 \pm 0.92$ | $15.12 \pm 1.87$ | $\mathbf{36.12}$ |
| DeepSeek-V2 -Lite | Frozen checkpoint | $35.42 \pm 2.11$ | $53.12 \pm 0.64$ | $13.93 \pm 0.91$ | $19.98 \pm 1.13$ | $\mathbf{37.20 \pm 3.73}$ | $\mathbf{69.20 \pm 0.87}$ | $19.40 \pm 1.77$ | $35.46$ |
| | ST-MoE | $47.95 \pm 2.21$ | $\mathbf{65.58 \pm 0.61}$ | $18.80 \pm 1.03$ | $25.48 \pm 1.23$ | $32.93 \pm 3.67$ | $68.12 \pm 0.88$ | $\mathbf{24.29 \pm 2.28}$ | $40.45$ |
| | **Ours** | $\mathbf{48.82 \pm 2.15}$ | $65.10 \pm 0.58$ | $\mathbf{19.34 \pm 1.00}$ | $\mathbf{26.21 \pm 1.19}$ | $33.55 \pm 3.58$ | $69.05 \pm 0.85$ | $23.94 \pm 2.21$ | $\mathbf{40.86}$ |

