# OpenReview forum: "$\phi$-Balancing for Mixture-of-Experts Training"
_ICML.cc/2026/Conference — ICML 2026 regular_

### Official Review · Reviewer_Deib · 2026-03-11

**Soundness:** 2
**Presentation:** 2
**Significance:** 3
**Originality:** 2
**Overall Recommendation:** 4
**Confidence:** 3

**Summary:**

The paper proposes $\phi$-balancing for Mixture-of-Experts training. The idea is to minimize a Schur-convex potential of the expected routing distribution. Via convex duality, the authors rewrite this as a min-max problem and derive a practical EMA-based mirror-descent update that yields a simple auxiliary loss.

**Compliance With Llm Reviewing Policy:**

Affirmed.

**Final Justification:**

My original review did not raise significant issues, and the rebuttals were satisfactory. I maintain my score.

**Key Questions For Authors:**

Not many questions at this point.

**Limitations:**

Yes.

**Strengths And Weaknesses:**

Soundness: The earlier parts appear sound, though I did not check the experimental setup in detail.

Presentation: Some parts could be improved (see below)

Significance/Originality: Yes. Load balancing as population objective optimization is a nice idea.

Some Strengths/Weaknesses:

- I like the paper's conceptual contribution. MoE load balancing as optimizing a population objective and progressing from Eq. (7) to Eq. (10), then to the EMA update in Algorithm 1 is quite clean.

- The resulting algorithm is simple and easy to add to existing MoE training pipelines.

- The evaluation is fairly extensive and appears consistent across multiple experiments.

- The pretraining data/setup, search ranges for tuned hyperparameters, evaluation protocol details for benchmarks like HumanEval, etc feel underspecified/not reproducible. This does impact soundness.

---

> ### Author Rebuttal · Authors · 2026-03-30
>
> We thank the reviewer for their thorough evaluation and constructive feedback. We will now address the reviewer’s concerns.
>
> We thank the reviewer for raising their concern about reproducibility. We believe that the current manuscript already provides a substantial amount of reproducibility detail, albeit somewhat distributed between the main text and Appendices C and D. In particular, the pretraining data and setup are specified in Section 4.1, with hyperparameter settings derived according to the scaling law of [1] in Appendix C, along with additional details on model architecture and training. The downstream setup and evaluation protocol are specified in Section 4.2, with complete detail regarding the 3 MoE backbones, 7 benchmarks, optimizers, hyperparameters, and tuning protocol. Accuracy on all benchmarks is computed by passing each model-generated solution together with the reference solution to a strong judge model (GPT-5.2), which determines functional correctness. For HumanEval specifically, because the benchmark itself does not provide a sufficient pool of training examples, we instead train on an example training set constructed from MBPP and evaluate zero-shot accuracy on HumanEval.
>
> We will include additional reproducibility details in a revision:
> - The C4 dataset [2] was used for Gemma pretraining.
> - For the hyperparameters listed in Table 5, our learning rate search employed a quasi-logarithmic grid spanning $1\times 10^{-5}$ to $1\times 10^{-1}$, with denser sampling in the $10^{-4}$ to $10^{-2}$ range where transformer models typically achieve optimal performance. The grid included standard decade values (e.g., 0.001, 0.01) as well as intermediate points within each logarithmic interval (e.g., 0.2, 0.3, 0.5, 0.8 scaled to each decade), totaling 24 distinct learning rate values. For the learning rate schedule, we systematically evaluated warmup ratios of 0, 0.05, 0.1, 0.2, 0.3, 0.4, 0.5 followed by cosine annealing decay.
>
> [1] Changxin Tian, Kunlong Chen, Jia Liu, Ziqi Liu, Zhiqiang Zhang, Jun Zhou. Towards Greater Leverage: Scaling Laws for Efficient Mixture-of-Experts Language Models. ICLR 2026.
>
> [2] Colin Raffel, Noam Shazeer, Adam Roberts, Katherine Lee, Sharan Narang, Michael Matena, Yanqi Zhou, Wei Li, Peter J. Liu. Exploring the Limits of Transfer Learning with a Unified Text-to-Text Transformer. Journal of Machine Learning Research, 21(140):1–67, 2020.

---

> > ### Author Rebuttal · Reviewer_Deib · 2026-03-31
> >
> > Thank you for your reply. I appreciate the authors clarifying they will include the reproducibility details in the revision. Generally speaking, it is good practice to include all details needed to reproduce the experiments in the paper appendix, which includes the learning rate search. My original review did not raise significant issues. I will keep my score.

---

### Official Review · Reviewer_dGUC · 2026-03-12

**Soundness:** 2
**Presentation:** 2
**Significance:** 3
**Originality:** 2
**Overall Recommendation:** 3
**Confidence:** 4

**Summary:**

This paper provides a quite new solution for how to best utilize the experts in MoE model. The core idea here is from the global perspective, not the local one. In fact, such idea is not so novel, but how to solve this is not easy. I think this paper provides a possible and interesting approach.

**Compliance With Llm Reviewing Policy:**

Affirmed.

**Key Questions For Authors:**

1. The potential function $\phi$ is important in this work, which is carefully chosen to be Schur-convex. But, unfortunately, why this potential can be defined in this way is NOT analyzed or discussed. This raises one following question, What if change the $\phi$ from Schur-convex to other forms? If the authors can present in-depth insights on this point, I would like to raise the score.

**Limitations:**

Yes

**Strengths And Weaknesses:**

1. The contribution of this work is NOT well summarized, even with inconsistency between the abstract and conclusion part. So, what is the $\phi$-Balancing in nature? what does this balancing brings to the routing in MoE, especially compared to not using this balancing? According my direct or simple understanding, the function or effect of this balancing is to change the distribution of of experts usage on all input tokens as a whole, which may be different from not using this balancing, although I am not so sure that I am right on this point. Just using "framework, loss, addition " sounds not adequate. If the authors can refine this, especially using mathematical forms, I would like to raise the point.

2. In this submission, "data" and "token" seem equivalent, I guess. I suggest the authors to clarify this, which is a little confusing. For example, "the entire data distribution" in line 141 pp 3 is NOT clearly defined.

3. The authors claim that "population-level expert balance" is the target of the proposed method, but unfortunately, here, the meaning of population is ambiguous. Does it mean the level of the data/token or the experts? Please clarify that.

4. The figures and corresponding discussions are hard to follow. For example, Figure 2 lies in pp 2 while such discussions lies in pp 5 with only "we systematically vary (i) the number of active parameters N, (ii) the number of
experts E; and (iii) the MoE granularity G (Figure 2).". I am confused by, what Figure 2 demonstrates and why the MoE granularity G connects with Figure 2. In fact, such separation, in page space and semantics, between figures and explanations occur for almost all these pairs.

5. One minor point is that, Figure 6 appears in section 4.1 before Figure 5 in section 4.2, which seems strange. Figure 4 holds the similar flaw.

---

> ### Author Rebuttal · Authors · 2026-03-30
>
> We thank the reviewer for their thorough evaluation and constructive feedback. We will now address the reviewer’s questions and concerns.
>
> 1. We thank the reviewer for raising this concern. We will work to revise the abstract, introduction, and conclusion to consistently and accurately summarize our contributions. The primary contribution of this work is to introduce $\phi$-balancing and provide theoretical and empirical evidence for its efficacy in balancing expert utilization for MoE models. In mathematical terms, this balancing is achieved by encouraging the population-level routing probabilities across experts to approach a **uniform distribution** via an auxiliary loss that is minimized at this distribution (Lemma 1). We observe that $\phi$-balancing provides a notable improvement in expert load balancing compared to standard baselines that do not use $\phi$-balancing. This is apparent in Figure 3, where the Gini coefficient (a measure of the inequality of a distribution) is significantly lower for $\phi$-balancing than for ST-MoE. We also verify that uniform expert usage is indeed a desirable property by observing superior performance on numerous ablation studies and pretraining/downstream tasks. We furthermore show that the mechanism of $\phi$-balancing can be implemented via maintaining a simple auxiliary loss, adding negligible overhead to existing pipelines.
>
> 2. In accordance with standard notation in the literature, we use $\mathcal{D}$ to denote the data-generating probability distribution (the “data distribution”) over the space of all possible inputs. In the context of MoE routing, this is specifically the empirical distribution of tokens induced by the training corpus, which is clear from, e.g., the notation $\boldsymbol{x}\sim\mathcal{D}$ in (6).
>
> 3. We use “population” and “population-level” to refer to expectations over the entire distribution $\mathcal{D}$ as opposed to the “local” mean of a mini-batch $\mathcal{B}$. Therefore, the meaning of “population” is at the level of tokens. We will clarify this in a revision.
>
> 4. We thank the reviewer for their constructive criticism. We agree that the current manuscript excessively separates several figures from their corresponding discussion, and we will improve the figure-text alignment in a revision. In particular, we will (i) strengthen the first mention of each figure to explain its main takeaways, (ii) move and restructure discussion to appear closer to the corresponding figure, and (iii) expand the captions to make them more self-contained. Regarding Figure 2 specifically, we will clarify that it summarizes our controlled-compute pretraining experiment across three distinct axes: active parameters $N$, expert count $E$, and MoE granularity $G$. We will explicitly state that the granularity $G$ controls how finely the FFN capacity is partitioned into experts while keeping total capacity and per-token compute fixed. This should make the connection between $G$ and Figure 2 immediately clear.
>
> 5. We thank the reviewer for pointing this out. We will correct this in a revision.
>
> Q1. In this work, the potential function $\phi$ is chosen to be Schur-convex so that the load balancing loss (7) is *uniquely minimized* at the uniform distribution (Lemma 1). The minimization problem (7) is reformulated as the min-max problem (10) via convex duality and solved using online mirror descent steps (12) (derived in Lemma 2). In particular, if $\phi$ is not Schur-convex, then Lemma 1 does not apply, and there is no guarantee that the minimum in (7) is uniquely achieved by the uniform distribution. It is possible that non-Schur-convex functions could work well in practice, but such choices of $\phi$ fall out of the scope of our theoretical analysis. We emphasize that requiring $\phi$ to be Schur-convex is not an overly restrictive assumption, as illustrated by, e.g., the examples given in Table 1.

---

### Official Review · Reviewer_8GnC · 2026-03-13

**Soundness:** 4
**Presentation:** 4
**Significance:** 2
**Originality:** 3
**Overall Recommendation:** 5
**Confidence:** 3

**Summary:**

This paper studies load balancing for Mixture-of-Experts (MoE) training. The main claim is that standard balancing methods are largely based on mini-batch statistics and therefore can be biased relative to the true population-level objective. The authors propose phi-balancing, which formulates balancing as minimizing a Schur-convex potential over the expected routing distribution, and derive a simple EMA-based online algorithm. Empirically, the method appears to improve routing stability, reduce capacity violations, and achieve better validation and downstream performance than Switch-style and related baselines.

**Compliance With Llm Reviewing Policy:**

Affirmed.

**Final Justification:**

The authors full resolved my concerns.

**Key Questions For Authors:**

Questions for the authors:
1. How sensitive is the method to the EMA step size and the auxiliary loss weight?
2. How much of the gain comes from EMA smoothing alone?
3. How much comes from replacing hard routing frequencies with soft routing probabilities?
4. Why does negative entropy work best empirically? Is there a stronger intuition beyond the current empirical observation?
5. Does stronger balancing ever hurt expert specialization or beneficial non-uniform routing on naturally skewed data?

**Limitations:**

yes

**Strengths And Weaknesses:**

Strengths:
1. Clear problem formulation. The paper makes a convincing case that batch-level balancing is not the same as population-level balancing. This is, in my opinion, the strongest part of the submission.
2. Simple and practical method. The proposed update seems lightweight and easy to integrate into existing MoE training pipelines.
3. Good empirical coverage. The paper reports results on both pretraining and downstream finetuning, and the reported gains appear reasonably consistent.
4. The mirror-map / phi view is conceptually neat and gives a unified way to think about multiple balancing objectives.

Weaknesses:
1. Limited novelty. While the formulation is cleaner than prior work, the practical algorithmic change seems modest. In effect, the method mainly replaces hard batch-level frequency statistics with a history-aware soft signal under a more principled objective. This makes the contribution feel more like a principled refinement than a fundamentally new routing/balancing mechanism.
2. Insufficient separation from recent EMA-based or loss-free balancing approaches. The paper explains the conceptual distinction, but the practical gap still feels somewhat narrow. I would like a more explicit comparison of what is truly new algorithmically versus what is a better theoretical interpretation of an existing design pattern.
3. Mechanism is not fully isolated. The paper argues that the gains come from optimizing a population-level objective, but the current experiments do not fully disentangle:
   - EMA smoothing,
   - soft probabilities vs. hard frequencies,
   - the specific choice of phi,
   - and the exact loss form.
4. The empirical section would be substantially stronger with more targeted ablations. Since the main thesis is about mini-batch bias, I expected stronger experiments directly probing that claim.

---

> ### Author Rebuttal · Authors · 2026-03-30
>
> We thank the reviewer for their thorough evaluation and constructive feedback. We will now address the reviewer’s questions and concerns.
>
> 1. We thank the reviewer for raising this concern, and we will revise the manuscript to clarify the separation between our proposed $\phi$-balancing and other methods. In contrast to prior work, we provide a **theoretically principled algorithm** for expert load balancing, improving upon heuristic algorithms such as ST-MoE. We believe that this theoretical insight, in and of itself, is already a valuable contribution. Moreover, while our algorithm may bear superficial resemblance to other methods by incorporating familiar techniques such as EMAs, there exist several critical differences. For instance, we replace the hard routing frequencies $f_e$ in ST-MoE with the EMA of soft routing probabilities under a mirror map $\nabla\phi$. Notably, this mapping converts between the primal and dual spaces and is performed *after* the EMA step. We refer to “Related methods” at the end of Section 3.1 for additional discussion.
>
> 2. See 1.
>
> 3. We provide the following ablation studies:
> - We ablate the choice of the EMA smoothing parameter $\eta$. In Figure 6, we see that EMA smoothing generally outperforms no EMA smoothing ($\eta=1$), with $\eta\in[0.6,0.7]$ being optimal.
> - We ablate EMA tracking of soft probabilities vs. EMA tracking of hard frequencies. In Table 3, we see that either choice yields comparable performance.
> - We ablate several choices of $\phi$. The results are summarized in Table 2, and negative entropy can be seen to be the best choice by a clear margin.
> - We ablate the exact loss form by comparing against ST-MoE and loss-free balancing on pretraining and downstream performance. The results are summarized in Figures 1-5 and Table 4.
>
> 4. We thank the reviewer for suggesting this ablation study, and we believe it significantly strengthens our central claim. The results are summarized below.
>
> **Table. Global batch size ablation on the 986M active-parameter Gemma-MoE setting ($E=16$, $A=2$).**
> Higher is better.
> | Method | Global Batch Size | Hellaswag ↑ | MMLU ↑ | C-eval ↑ |
> |---|---:|---:|---:|---:|
> | ST-MoE | 32 | 62.82 | 41.96 | 42.58 |
> | ST-MoE | 128 | 63.14 | 42.37 | 43.24 |
> | ST-MoE | 512 | 63.34 | 42.74 | 43.87 |
> | Loss-Free | 32 | 62.38 | 41.58 | 42.87 |
> | Loss-Free | 128 | 62.73 | 42.03 | 43.46 |
> | Loss-Free | 512 | 63.05 | 42.46 | 44.00 |
> | Ours ($\phi$-balancing) | 32 | 63.46 | 42.88 | 43.96 |
> | Ours ($\phi$-balancing) | 128 | 63.60 | 43.02 | 44.18 |
> | Ours ($\phi$-balancing) | 512 | **63.70** | **43.18** | **44.36** |
>
> Q1. Regarding hyperparameter sensitivity, we explicitly conducted ablation studies on both the EMA step size $\eta$ and the auxiliary loss coefficient $\alpha$. For the EMA step size $\eta$, we refer to the above discussion and Figure 6. For the auxiliary loss coefficient, Figure 7 in the Appendix reports a **4×4 sensitivity sweep** over the learning rate $\gamma$ and the balancing loss coefficient $\alpha$. Across different choices of $\alpha$, validation accuracy remains robust, while the Gini coefficient consistently decreases as $\alpha$ increases, indicating stronger balancing. Overall, these ablation studies indicate that $\phi$-balancing does not exhibit excessive hyperparameter sensitivity.
>
> Q2. From Figure 6, we can see that for values of $\eta$ close to 1, corresponding to low levels of EMA smoothing, the EMA becomes too close to noisy single-batch statistics, resulting in significant performance degradation.
>
> Q3. See Table 3.
>
> Q4. We ask the reviewer to clarify what is meant by “naturally skewed data”. In our post-training experiments (Table 4), we evaluate $\phi$-balancing on datasets that are highly domain specific (code and math), and the average performance is consistently higher than that of ST-MoE.
>
> Q5. We thank the reviewer for raising this important question. We do not necessarily claim that negative entropy is the absolute optimum among all possible choices of $\phi$, but we can share our thoughts on why it performs so effectively in practice compared to other variants. In the $\phi$-balancing framework, the router tries to minimize a loss that pairs the current batch’s routing probabilities $\mathbf{p}$ with a dual variable $\mathbf{q}$, which can be interpreted as the vector of “prices” of using expert $e$ based on its historical usage $\mathbf{m}_e$. The choice of $\phi$ determines exactly how $\mathbf{m}$ translates into $\mathbf{q}$ via (12), which is $\mathbf{q}=\log\mathbf{m}+1$ for the negative entropy potential $\phi(\mathbf{m})=\sum\mathbf{m}_e\log\mathbf{m}_e$. If an expert is “starving” and its historical usage $\mathbf{m}_e$ approaches $0$, its price $\mathbf{q}_e$ rapidly drops toward $-\infty$, acting as a massive “discount” and creating an overwhelming incentive for the router to send more tokens to that expert. We believe that deeper exploration of this question is an exciting direction for future work.

---

> > ### Author Rebuttal · Reviewer_8GnC · 2026-04-06
> >
> > The authors fully resolved my concerns. I raised my score.

---

### Official Review · Reviewer_NWQJ · 2026-03-13

**Soundness:** 3
**Presentation:** 4
**Significance:** 3
**Originality:** 3
**Overall Recommendation:** 5
**Confidence:** 3

**Summary:**

The paper introduces $\phi$-balance loss for MOE routers to enforce expert load-balancing.  The proposed loss is a population-level expert balance by minimizing a Schur-convex potential of the expected routing distribution. To enable mini-batch training while maintaining the population statistics, EMA-based routing probabilities are derived based on convex duality and mirror descent. Experiments on pre-training and fine-tuning tasks demonstrate the efficacy of the proposed method.

**Compliance With Llm Reviewing Policy:**

Affirmed.

**Final Justification:**

The authors have addressed my concerns. I maintain my score.

**Key Questions For Authors:**

1. In Figure 5, the proposed method has mildly uneven expert loads compared to ST-MoE on downstream fine-tuning tasks, which seems inconsistent with the main claim that their method should be more load-balancing due to population-level statistics.
2. Why does the proposed method exhibit sharper domain-to-expert preferences?

**Limitations:**

The authors have not discussed the limitations of their work. It may be worthwhile to discuss any potential issues with the proposed method or future extensions.

**Strengths And Weaknesses:**

Strengths:
1. The paper addresses an important problem for MOE training. The proposed method is simple yet solid, and the derivation of the EMA-based solution is elegant.
2. The experiments are sufficient and demonstrate promising results.
3. The writing is clear and easy to follow.

I have some concerns. (1) In Figure 5, the authors mention that their proposed method has mildly uneven expert loads compared to ST-MoE on downstream fine-tuning tasks, which seems inconsistent with the main claim that their method should be more load-balancing due to population-level statistics.  (2) It would be better to include more explanation or analysis of why the proposed method exhibits sharper domain-to-expert preferences.

---

> ### Author Rebuttal · Authors · 2026-03-30
>
> We thank the reviewer for their thorough evaluation and constructive feedback. We will now address the reviewer’s questions and concerns.
>
> Q1. We apologize for the confusion, and we will clarify this in a revision. Figure 5 is in fact consistent with our main claim, but we did not state clearly enough that the visualization is for specific layers over **a single mini-batch**. $\phi$-balancing encourages the *population-level* mean routing distribution to approach a uniform distribution, whereas Figure 5 visualizes *domain-specific* distributions at specific layers. Crucially, a uniform distribution does **not** require every conditional marginal to also be uniform. In other words, $\phi$-balancing ensures that experts are balanced in aggregate over the whole dataset, but not necessarily within every mini-batch, every domain, or every layer. In Section 3.1, we also explicitly note that batchwise balancing is biased because it artificially forces the router to balance every individual mini-batch rather than the global distribution. This explains the perceived imbalance of the conditional distributions in Figure 5.
>
> Q2. Figure 5 actually suggests that the $\phi$-balancing router learns more meaningful expert roles compared to ST-MoE. Standard load-balancing losses, such as that of ST-MoE, are calculated using token allocation statistics strictly within each mini-batch. However, due to the limited size of mini-batches, they are often dominated by a single domain (e.g., arXiv papers or GitHub code) and rarely represent the entire dataset’s diversity. Since ST-MoE encourages the router to distribute the tokens of each mini-batch uniformly across all experts, the result is that every expert is forced to learn every domain, suppressing expert specialization. In contrast, $\phi$-balancing accumulates routing statistics across many mini-batches over time using an EMA, which acts as a proxy for the *global* data distribution. Because the strict mini-batch constraints are relaxed, $\phi$-balancing permits temporary or domain-specific load imbalances. If a batch consists mostly of, e.g., GitHub code, the router is free to send the majority of those tokens to a few specific “code experts” without incurring a massive immediate penalty, unlike ST-MoE. This is why $\phi$-balancing exhibits meaningful expert roles and sharper domain-to-expert preferences.
>
> Regarding limitations and future directions, Reviewer 8GnC raises a good question about the optimal choice of $\phi$, and we will include additional discussion in a revision.

---

> > ### Author Rebuttal · Reviewer_NWQJ · 2026-04-03
> >
> > Thanks for addressing my concerns. I maintain my score.

---

### Decision · Program_Chairs · 2026-04-30

**Decision:**

Accept (regular)

**Comment:**

This paper proposes $\phi$-balancing, a principled load-balancing approach for MoE training. By leveraging a Schur-convex potential, the method formulates a dual objective that can be efficiently optimized via mirror descent using an EMA of loads. Empirically, the approach consistently outperforms existing load-balancing strategies without introducing noticeable computational overhead.

Overall, the reviewers find the paper to be of high quality, with strong empirical results. In contrast to prior heuristic methods that rely on mini-batch statistics, the proposed method is grounded in a population-level optimization perspective while remaining practical for large-scale training. Reviewer dGUC raised several concerns, primarily regarding presentation clarity (notation and figures), which I believe can be adequately addressed in the camera-ready version.

One additional note concerns a reported hallucinated reference:
Thai, H., Vu, N., Phan, M., Ly, T., Dinh, T., Nguyen, T., and Ho, N. Shape-adapting gated experts: Dynamic expert routing for colonoscopic lesion segmentation. arXiv preprint arXiv:2502.13322, 2025.
The author names are not properly abbreviated, the title is missing the word “SAGE,” and the arXiv identifier appears to be incorrect. According to ICML policy, submissions with two or more hallucinated references are subject to desk rejection. While only one such issue has been identified here and thus does not affect the current decision, **this reference must be corrected in the final version**.